**A mosaic of phytoplankton responses across Patagonia, the southeast Pacific and**
**southwest Atlantic Oceans to ash deposition and trace metal release from the Calbuco**
**volcanic eruption in 2015**
Maximiliano J. Vergara-Jara[1,2], Mark J. Hopwood[3*], Thomas J. Browning[3], Insa Rapp[4],
Rodrigo Torres[2,5], Brian Reid[5], Eric P. Achterberg[3], José Luis Iriarte[2,6].
[1]Programa de Doctorado en Ciencias de la Acuicultura, Universidad Austral de Chile, Puerto
Montt, Chile.
[2]Instituto de Acuicultura & Centro de Investigación Dinámica de Ecosistemas Marinos de
Altas Latitudes - IDEAL, Universidad Austral de Chile, Puerto Montt, Chile.
[3]GEOMAR, Helmholtz Centre for Ocean Research, 24148 Kiel, Germany.
[4]Department of Biology, Dalhousie University, Halifax, Nova Scotia, Canada
[5]Centro de Investigación en Ecosistemas de la Patagonia (CIEP), Coyhaique, Chile.
[6]COPAS-Sur Austral, Centro de Investigación Oceanográfica en el Pacífico Sur-Oriental
(COPAS), Universidad de Concepción, Concepción, Chile.
Key words: volcanic ash, iron, Fe(II), phytoplankton, carbonate chemistry, Reloncaví Fjord
Corresponding author*: mhopwood@geomar.de

## Abstract

Following the eruption of the Calbuco volcano in April 2015, an extensive ash plume spread across northern Patagonia and into the southeast Pacific and southwest Atlantic Oceans. Here we report on field surveys conducted in the coastal region receiving the highest ash load following the eruption (Reloncaví Fjord). The fortuitous location of a long-term monitoring station in Reloncaví Fjord provided data to evaluate inshore phytoplankton bloom dynamics and carbonate chemistry during April-May 2015. Satellite derived chlorophyll-a measurements over the ocean regions affected by the ash plume in May 2015 were obtained to determine the spatial-temporal gradients in offshore phytoplankton response to ash. Additionally, leaching experiments were performed to quantify the release from ash into solution of total alkalinity, trace elements (dissolved Fe, Mn, Pb, Co, Cu, Ni and Cd) and major ions ($F^-$, $Cl^-$, $SO_4^{2-}$, $NO_3^-$, $Li^+$, $Na^+$, $NH_4^+$, $K^+$, $Mg^{2+}$, $Ca^{2+}$). Within Reloncaví Fjord, integrated peak diatom abundances during the May 2015 austral bloom were approximately 2-4 times higher than usual (up to $1.4 \times 10^{11}$ cells $m^{-2}$, integrated to 15 m depth), with the bloom intensity perhaps moderated due to high ash loadings in the two weeks following the eruption. Any mechanistic link between ash deposition and the Reloncaví diatom bloom can however only be speculated on due to the lack of data immediately preceding and following the eruption. In the offshore southeast Pacific, a short duration phytoplankton bloom corresponded closely in space and time to the maximum observed ash plume, potentially in response to Fe-fertilization of a region where phytoplankton growth is typically Fe-limited at this time of year. Conversely, no clear fertilization on the same time-scale was found in the area subject to an ash plume over the southwest Atlantic where the availability of fixed nitrogen is thought to limit phytoplankton growth. This was consistent with no significant release of fixed nitrogen ($NO_x$ or $NH_4$) from Calbuco ash.

62

In addition to release of nanomolar concentrations of dissolved Fe from ash suspended in

seawater, it was observed that low loadings ($< 5$ mg L$^{-1}$) of ash were an unusually prolific

source of Fe(II) into chilled seawater (up to 1.0 µmol Fe g$^{-1}$), producing a pulse of Fe(II)

typically released mainly during the first minute after addition to seawater. This release

would not be detected, either as Fe(II) or dissolved Fe, following standard leaching protocols

at room temperature. A pulse of Fe(II) release upon addition of Calbuco ash to seawater made

it an unusually efficient dissolved Fe source. The fraction of dissolved Fe released as Fe(II)

from Calbuco ash (~18-38%) was roughly comparable to literature values for Fe released

into seawater from aerosols collected over the Pacific Ocean following long range

atmospheric transport.

## 1. Introduction

Volcanic ash has long been considered a large, intermittent source of trace metals to the ocean (Frogner et al., 2001; Sarmiento, 1993; Watson, 1997) and its deposition is now deemed a sporadic generally low-macronutrient, high-micronutrient supply mechanism (Ayris and Delmelle, 2012; Jones and Gislason, 2008; Lin et al., 2011). As volcanic ash can be a regionally significant source of allochthonous inorganic material to affected water bodies, volcanic eruptions have the potential to dramatically change light availability, the carbonate system, properties of sinking particles and ecosystem dynamics (Hoffmann et al., 2012; Newcomb and Flagg, 1983; Stewart et al., 2006). Surveys directly underneath the ash plume from the 2010 eruption of Eyjafjallajökull (Iceland) over the North Atlantic found, among other biogeochemical perturbations, high dissolved Fe (dFe) concentrations of up to 10 nM in affected surface seawater (Achterberg et al., 2013) which could potentially result in enhanced primary production. The greatest potential positive effect of ash deposition on marine productivity would generally be expected in high-nitrate, low-chlorophyll (HNLC) areas of the ocean (Hamme et al., 2010; Mélançon et al., 2014), where low Fe concentrations are a major factor limiting primary production (Martin et al., 1990; Moore et al., 2013). Special interest is therefore placed on the ability of volcanic ash to release dFe, and other bio-essential trace metals such as Mn (Achterberg et al., 2013; Browning et al., 2014; Hoffmann et al., 2012), into seawater. In contrast, apart from inducing light limitation, there are several adverse effects of ash deposition on aquatic organisms. These include metal toxicity (Ermolin et al., 2018), particularly under high dust loading (Hoffmann et al., 2012), and the ingestion of ash particles by filter feeders, phagotrophic organisms or fish (Newcomb and Flagg, 1983; Wolinski et al., 2013). Transient shifts to low pH have also been reported in some, but not all, ash leaching experiments and in some freshwater bodies following

intense ash deposition events, suggesting that significant ash deposition on weakly buffered
aquatic environments can also impact and perturb their carbonate system (Duggen et al.,
2010; Jones and Gislason, 2008; Newcomb and Flagg, 1983). The greatest negative impact
of ash on primary producers would therefore be expected closest to the source where the ash
loading is highest and in areas where macronutrients or light, rather than trace elements, limit
primary production.

In contrast to the 2010 Eyjafjallajökull plume over the North Atlantic, the 2015 ash plume
over the region from the Calbuco eruption (northern Patagonia, Chile) was predominantly
deposited over an inshore and coastal region (Romero et al., 2016) (Fig. 1). This led to visible
high ash loadings in affected surface waters in the weeks after the eruption (Fig. 2), providing
a case study for a concentrated ash deposition event in a coastal system; Reloncaví Fjord,
which is the northernmost fjord of Patagonia. It receives the direct discharge from three major
rivers, creating a highly stratified and productive fjord system in terms of both phytoplankton
biomass and aquaculture production of mussels (González et al., 2010; Molinet et al., 2017;
Yevenes et al., 2019). Here we combine in situ observations from moored arrays which were
fortuitously deployed in Reloncaví Fjord (Vergara-Jara et al., 2019), with satellite-derived
chlorophyll data for offshore regions subject to ash deposition, and leaching experiments
carried out on ash collected from the fjord region, to investigate the inorganic consequences
of ash addition to natural waters. We thereby evaluate the potential positive and negative
effects of ash from the 2015 Calbuco eruption on marine phytoplankton in three geographical
regions; Reloncaví Fjord and the areas of the SE Pacific and SW Atlantic Oceans beneath
the most intense ash plume.

2. **Materials and methods**
2.1. *Study area*
The Calbuco volcano (Fig. 1) is located in a region with large freshwater reservoirs and in
close proximity to Reloncaví Fjord. The predominant bedrock type is andesite (López-
Escobar et al., 1995). Reloncaví Fjord is 55 km long and receives freshwater from 3 main
rivers, the Puelo, Petrohué, and Cochamó, with mean stream flows of 650 m$^3$ s$^{-1}$, 350 m$^3$ s$^{-1}$
and 100 m$^3$ s$^{-1}$, respectively (León-Muñoz et al., 2013). River discharge strongly influences
seasonal patterns of primary production across the region, supplying silicic acid and strongly
stratifying the water column (Castillo et al., 2016; González et al., 2010; Torres et al., 2014).
Seasonal changes in light availability rather than macronutrient supply are thought to control
marine primary production across the Reloncaví region with high marine primary production
(>1 g C m$^{-2}$ day$^{-1}$) throughout austral spring, summer and early autumn (González et al.,

133    2010).

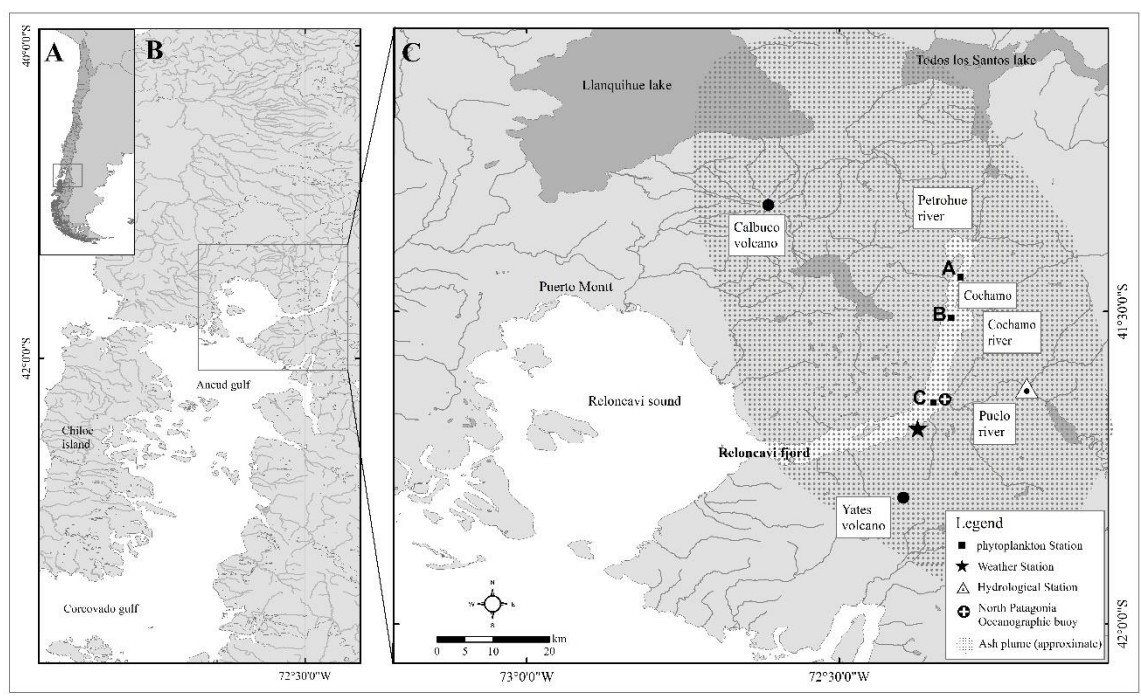


Figure 1. The Calbuco region showing the location of Reloncaví Fjord, 3 major rivers
(Petrohué, Cochamó and Puelo) discharging into the fjord, the 3 stations (black squares; A,
B and C) used to assess changes in phytoplankton abundance following the eruption, a
hydrological station that monitors Puelo river flow, a weather station and the location of a
long-term mooring within the fjord. The approximate extent of the ash plume in the week
following the first eruption is illustrated, as estimated in technical reports issued by the
Servicio Nacional de Geología y Minería (Chile).

On 22 April 2015 the Calbuco volcano erupted after 54 years of dormancy. Two major
eruption pulses lasted <2 hours on 22 April and 6 hours on 23 April, releasing a total volume
of 0.27 km$^3$ ash which was projected up to 20 km height above sea level (Van Eaton et al.,
2016; Romero et al., 2016). Ash layers of several cm thick were deposited mainly to the NE
of the volcano in subsequent days (Romero et al., 2016). A smaller eruption occurred on 30
April projecting ash 4-5 km above sea level which was then mainly deposited south of the
volcano. Smaller volumes of ash were released semi-continuously for three weeks after the
main eruption, leading to intermittent ash deposition events. Fortuitously, as part of a long-
term deployment, an ocean acidification buoy in the middle of Reloncaví Fjord (Vergara-
Jara et al., 2019) and an associated meteorological station close to the volcano (Fig. 1) were
well placed to assess the impact of ash deposition immediately after the eruption. To
complement data from these facilities, after the regional evacuation order was removed,
weekly sampling campaigns were conducted in the fjord commencing one week after the
eruption. The Chilean Geological-mining Survey (Servicio Nacional de Geología y Minería,
SERNAGEOMIN) produced daily technical reports including the estimated area of ash
dispersion        (http://sitiohistorico.sernageomin.cl/volcan.php?pagina=4&iId=3).        This
information was used to create a reference aerial extent of ash deposition for the week after
the eruption (Fig. 1, C). This approximation represents a full week of coverage for this
dynamic feature.

**2.2. Ash samples – trace metal leaching experiments**
On 6 May (2015, Cochamó, Chile, approximately 30 km from the volcano) after the third,
and smallest, eruptive pulse of ash from the Calbuco volcano (Fig. 2, A), and with the volcano
still emitting material, ash was collected using a plastic tray wrapped with plastic sheeting
($40 \times 94$ cm). The plasticware was left outside for 24 hours until sufficient ash (~500 g) was
collected to provide a bulk sample. Ambient weather over the period of ash collection, and
the preceding day, was dry (no precipitation). The collected ash was double sealed in low
density polyethylene (LDPE) plastic bags and stored in the dark. A sub-sample was analyzed
for particle size using a Mastersizer 2000 at The University of Chile.
Ash may affect in situ phytoplankton dynamics in several ways, for example via moderating
the carbonate system, macronutrient availability and/or micronutrient availability. As
micronutrient (e.g. Fe and Mn) availability is expected to be the main chemical mechanism
via which phytoplankton dynamics in the offshore marine environment could be affected, we
primarily focus our investigation on the release of dissolved trace metals from ash in
seawater. Yet to rule out other potential affects, we also conduct complementary leaches to
assess the significance of changes to total alkalinity and macronutrient availability (Table 1).
For trace metal leaches, a variety of methods have been used in the literature (Duggen et al.,
2010; Witham et al., 2005) depending on the purpose of specific studies. De-ionized water
leaches with ash loadings that are high in an offshore environmental context are preferable
for intercomparison studies. The trace metals released under such conditions are however
difficult to compare quantitatively to metal exchange processes in the ambient marine
environment, especially for elements such as Fe where solubility is strongly influenced by
pH, salinity and the nature of dissolved organic carbon present (Baker and Croot, 2010). For
prior work conducted specifically using volcanic ash in seawater, 3 main methods have been
employed: suspension experiments followed by analysis of the leachate, flow-through
reactors, and continuous voltammetric determination of dFe concentrations in situ during
suspension experiments (Sup. Table 1). The most commonly used ash:solute ratio in prior
seawater experiments is 1:400 (g:mL), with leach lengths varying from 15 minutes to 24
hours (Sup. Table 1). Conversely, incubation experiments designed to test the response of
marine phytoplankton to ash deposition have used lower ash:solute ratios of  1:400 to $1:10^7$
which are based on estimates of the ash loading expected to be mixed within the offshore
surface mixed layer underneath ash plumes (Browning et al., 2014; Hoffmann et al., 2012).
Existing data suggests that ash:solute ratio is not a major factor in determining the release
behavior of Fe from ash, however this is acknowledged to be difficult to assess due to other
differences between experimental setups used to date (Duggen et al., 2010). Both the age of
particles since collection and the organic carbon content of seawater are however known to
be critical factors influencing the exchange of Fe, and other trace elements, following any
aerosol deposition into seawater (Baker and Croot, 2010; Duggen et al., 2010). Whilst UV-
treatment of seawater has been used in some experiments (to remove a large part of any
natural organic ligands present, Duggen et al., 2007; Jones and Gislason, 2008), and a strong
synthetic organic ligand added in others (to impede dissolved Fe precipitation, Duggen et al.,
2007; Olgun et al., 2011; Simonella et al., 2015), to improve reproducibility and
standardisation, these steps are not well suited specifically for investigating the release of
Fe(II) from ash. Herein we therefore adopt ash:solute ratios comparable to the lower end of
the range used in leaching experiments and comparable to the range used in incubation
experiments. Seawater was used after prolonged storage in the dark (to reduce biological
activity to low background levels) and without UV treatment (to maintain an environmentally
relevant level of natural organic material in solution). A short leaching time (10 minutes +
filtration) was adopted to minimize bottle effects and recognising that most prior work
suggests a large fraction of Fe release occurs on short timescales (minutes), followed by more
gradual changes on timescales of hours to days (Duggen et al., 2007; Frogner et al., 2001;
Jones and Gislason, 2008).
A variety of leaches were conducted in de-ionized water, brackish (fjord) water or offshore
South Atlantic seawater (Table 1) with the choice of leaching conditions based on the
expected environmental significance in different water masses. Offshore oligotrophic
seawater for incubation experiments was collected from an underway transect of the mid-
South Atlantic (across 40° S) using a towfish and trace metal clean tubing in a 1 $m^3$ high
density polyethylene tank which had been pre-rinsed with 1 M HCl. This water was stored
in the dark for >12 months prior to use in leaching experiments and was filtered
(AcroPak1000 capsule 0.8/0.2 µm filters) when subsampling a batch for use in all leaching
experiments. All labware for trace metal leaching experiments was pre-cleaned with Mucasol
and 1 M HCl. 125 ml LDPE bottles (Nalgene) for trace metal leach experiments were pre-
cleaned using a 3-stage procedure with three de-ionized water (Milli-Q, Millipore,
conductivity 18.2 MΩ $cm^{-1}$) rinses after each stage (3 days in Mucasol, 1 week in 1 M HCl,
1 week in 1 M $HNO_3$).
Leach experiments were conducted by adding a pre-weighed mass of ash into 100 ml South
Atlantic Seawater, gently mixing the suspension for 10 minutes, and then syringe filtering
the suspension (0.2 µm, polyvinylidene fluoride, Millipore). Eight different ash loadings
from 2-50 mg L$^{-1}$ were used, selected to be environmentally relevant and comparable to prior
incubation experiments, with each treatment run in triplicate. Samples for dissolved trace
metals (Fe, Cd, Pb, Ni, Cu, Co and Mn) were acidified within 1 day of collection by the
addition of 140 µL concentrated HCl (UPA grade, ROMIL) and analysed by inductively
coupled plasma mass spectroscopy following preconcentration exactly as per Rapp et al.,

236  (2017).

Leach experiments specifically to measure Fe(II) release were conducted in a similar manner
but in cold seawater with continuous in-line analysis (5-7°C see Sup. Table 2) due to the
rapid oxidation rate of Fe(II) at room temperature (~21°C), which makes accurate
measurement of Fe(II) concentrations challenging (Millero et al., 1987). For these
experiments, a pre-weighed mass of ash was added to 250 ml South Atlantic seawater and
manually shaken for approximately one minute, using an expanded loading range from 0.2-
4000 mg L$^{-1}$. Fe(II) was measured via flow injection analysis using luminol
chemiluminescence (Jones et al., 2013) without pre-concentration or filtration. The inflow
line feeding the flow injection apparatus was positioned inside the ash suspension
immediately after mixing and measurements begun thereafter at 2 minutes resolution.
Reported mean values (± standard deviation) are determined from the Fe(II) concentrations
measured 2-30 minutes after adding ash into solution. Calibrations were run daily using
standard additions of 0.2-10 nM Fe(II) to aged South Atlantic seawater at the same
temperature with integrated peak area used to construct calibration curves. Following each
leaching experiment the apparatus was rinsed with 0.1 M HCl (reagent grade) followed by
flushing with de-ionized water to ensure the removal of ash particles. Blank measurements
before/after Fe(II) measurements from experiments with different ash loadings verified that
there was no discernable interference from ash particles in the Fe(II) flow-through
measurements. Fe(II) leaches were conducted 2 weeks, 4 months and 9 months after the
eruption. Fe(II) leaches 2 weeks after the eruption were run for 30 minutes. Fe(II) leaches
after 4 or 9 months were run for 1 hour to further investigate the temporal development of
Fe(II) concentration. The trace metal leach experiments (above) were conducted at the same
time as the first Fe(II) incubation experiments (2 weeks after ash collection).
For trace metal leaches, the initial (mean $\pm$ standard deviation) dissolved trace metal
concentrations were deducted from the final concentrations, in order to calculate the net
change as a result of ash addition. For Fe(II) measurements, background levels of Fe(II) were
below detection (<0.1 nM) and so no deduction was made.
**2.3 Ash samples – de-ionized and brackish water leaching experiments**
Fresh brackish sub-surface water from the Patagonia study region was obtained from the
Aysén Fjord, at Ensenada Baja (45º21'S: 72º40'W, salinity 16.3), close to the Coyhaique
laboratory (Aysén region, Chile) and free from the influence of ash from the 2015 eruption.
The oceanographic conditions in these waters are similar to the adjacent Reloncaví fjord
(Cáceres et al., 2002). De-ionized water, along with the Aysén fjord brackish water, were
used for leaching experiments using two size fractions of ash following the general
recommendations of Duggen et al., (2010) and Witham et al., (2005) to consider the effects
of different size fractions and leachates. Leaches were conducted in 50 ml LDPE bottles filled
with either 40 ml brackish or DI-water with 4 replicates of each treatment. Bottles were
incubated inside a mixer at room temperature after the addition of 0.18 g ash, using two ash
size fractions (<63 µm and 250-1000 µm) which were separated using sieves (ASTM e-11
specification, W.S. Tyler). The mass distribution of the ash as determined by sieving was
4.54% >2360 µm; 6.85% <2360 µm and >1000 µm; 31.12% <1000 µm and >250 µm;
24.14% <250 µm and >125 µm; 18.04% <125 µm and >63 µm; 15.31% <63 µm. The
dominant size fraction by mass was thereby the 250-1000 µm fraction which was analyzed
in addition to the finest fraction (<63 µm) with the greatest surface area to mass ratio. The
sampling times were at time zero (defined as just after the addition of the ash and a few
minutes of mixing), 2 h and 24 h later. Leaching experiments conducted with brackish water
were analyzed for total alkalinity ($A_T$) via a potentiometric titration using reference standards
(Haraldsson et al., 1997) ensuring a reproducibility of <2 µmol/kg. For the de-ionized water
leaching experiment, $A_T$ was analyzed by titration of unfiltered 5 ml subsamples to a pH 4.5
endpoint (Bromocresol Green/Methyl Red) using a Dosimat (Metrohm Inc) and 0.02 N
$H_2SO_4$ titrant. Alkalinity was calculated as $CaCO_3$ equivalents following APHA (American
Public Health Association) 2005-Methods 2320 (2320 Alkalinity, titration method).
Additional 5 ml subsamples were filtered, stored at 4°C and analyzed within 3 days for major
ions ($F^-$, $Cl^-$, $SO_4^{2-}$, $NO_3^-$, $Li^+$, $Na^+$, $NH_4^+$, $K^+$, $Mg^{2+}$, $Ca^{2+}$) using a DionexTM 5000 Ion
Chromatography system with Eluent Generation (APHA). All measurements were then
corrected for initial water concentrations prior to ash addition. Saturation indices for species
in solution following leaching from <63 µm ash particles were obtained from the MINTEQ
3.1. IAP Ion Activity Product chemical equilibrium model (see Sup. Table 6).
Table 1. Summary of different leaching experiments and samples.

| Ash/ particle source | De-ionized water leaches | Brackish (fjord) water | South Atlantic seawater | Number of replicates |
|---|---|---|---|---|
| Calbuco ash, sieved <63 μm | Total alkalinity, ion and macronutrients | Total alkalinity | - | 4 |
| Calbuco ash, sieved 250-1000 μm | Total alkalinity, ion and macronutrients | Total alkalinity | - | 4 |
| Calbuco ash, unsieved | - | - | Trace metals, Fe(II) | 3 for trace elements, 1 time series for Fe(II) |

**2.4 Environmental data – continuous Reloncaví Fjord monitoring**

High temporal resolution (hourly) in situ measurements were taken in the Reloncaví fjord (Fig. 1 C, North Patagonia Oceanographic Buoy) at 3 m depth using SAMI sensors that measured spectrophotometric $CO_2$ and pH (DeGrandpre et al., 1995; Seidel et al., 2008) (Sunburst Sensors, LLC), and an SBE 37 MicroCAT CTD-ODO (SeaBird Electronics) for temperature, conductivity, depth and dissolved $O_2$, as per Vergara-Jara et al., (2019). Sensor maintenance and quality control is described by Vergara-Jara et al., (2019). The error in $p$CO$_2$ concentrations is estimated to be at most 5% which arises mainly due to a non-linear sensor response and reduced sensitivity at high $p$CO$_2$ levels >1500 ppm (DeGrandpre et al., 1999). The SAMI-pH instruments used an accuracy test instead of a calibration procedure (Seidel et al., 2008). With the broad pH and salinity range found in the fjord, pH values are subject to a maximum error of ±0.02 (Mosley et al., 2004).

A meteorological station (HOBO-U30, Fig. 1) measured air temperature, solar radiation,

wind speed and direction, rainfall, and barometric pressure every 5 minutes. Puelo River
streamflow was obtained from the Carrera Basilio hydrological station (Fig. 1), run by
Dirección General de Aguas de Chile (http://snia.dga.cl/BNAConsultas/reportes).
**2.5 Field surveys in Reloncaví Fjord post eruption**
During May 2015, weekly field campaigns were undertaken in the Reloncaví Fjord.
Phytoplankton samples were collected at 3 depths (1, 5 and 10 m) for taxonomic
characterization and abundance determination at 3 stations (A, B and C; Fig. 1) using a 5 L
Go-Flo bottle. Samples for cell-counts were stored in clear plastic bottles (300 mL) and
preserved in a Lugol iodine solution. From each sample, a 10 mL subsample was placed in a
sedimentation chamber and left to settle for 16 hr. The complete chamber bottom was
scanned at 200× to enumerate the organisms and the result was expressed as number of
phytoplankton cells per L of seawater (Hasle, 1978). Phytoplankton were identified to genus
or species level, when possible, and divided into diatoms and dinoflagellates. Samples were
analyzed using an Olympus CKX41 inverted phase contrast microscope and the Utermöhl
method (Utermöhl, 1958). The phytoplankton community composition was then statistically
analyzed in R (RStudio V 1.2.5033) using general linear models in order to find statistically
significant differences between dates and group abundances. Additionally, as part of a long-
term monitoring program at station C (Fig. 1), chlorophyll-a samples were retained from 6
depths (1, 3, 5, 7, 10 and 15 m) on 6 occasions during March-May 2015. Chlorophyll-a was
determined by fluorometry after filtering 250 ml of sampled water through GFF filters
(Whatman) as per Welschmeyer (1994). Two additional profiles close to Station C were
obtained from Yevenes et al., (2019). Integrated chlorophyll-a (mg $m^{-2}$) and diatom
abundance (cells $m^{-2}$) were determined to 15 m depth. Chlorophyll-a within Reloncaví Fjord
is invariably concentrated in the upper ~10 m (González et al., 2010; Yevenes et al., 2019)
and thus, for comparison to prior reported data integrated to 10 m, only a small difference is
anticipated. For all profiles considered herein, there is a 20% difference between integrating
to 10 m or 15 m depth.
**2.6 Satellite data**
Daily, 4 km resolution chlorophyll-a images from the MODIS Aqua sensor (OCI algorithm;
Hu  et  al.,  2012)  were  downloaded  from  the  NASA  Ocean  Color  website
(https://oceancolor.gsfc.nasa.gov) for the period 4 April 2015–2 May 2015. As the UV
Aerosol Index largely reflects strongly UV-absorbing (dust) aerosols (Torres et al., 2007),
this was used as a proxy for the spatial extent and loading of the ash plume. The UV aerosol
index product from the Ozone Monitoring Instrument (OMI) on the EOS-Aura was
downloaded for the same time period. Daily images were composited into 5-day mean
averages.
**3. Results**
**3.1 In situ observations**
The Calbuco ash plume reached up to 20 km height and was dispersed hundreds of kilometers
across Patagonia and the Pacific and Atlantic Oceans (Fig. 2) (Van Eaton et al., 2016;
Reckziegel et al., 2016; Romero et al., 2016).  The ash loading in water bodies near the cone
was visually observed to be high, especially near the Petrohué river catchment that drains
into the head of the Reloncaví fjord. This ash loading into the fjord was clearly visible on 6
May 2015 when ash samples were collected for leaching experiments (Fig. 2).

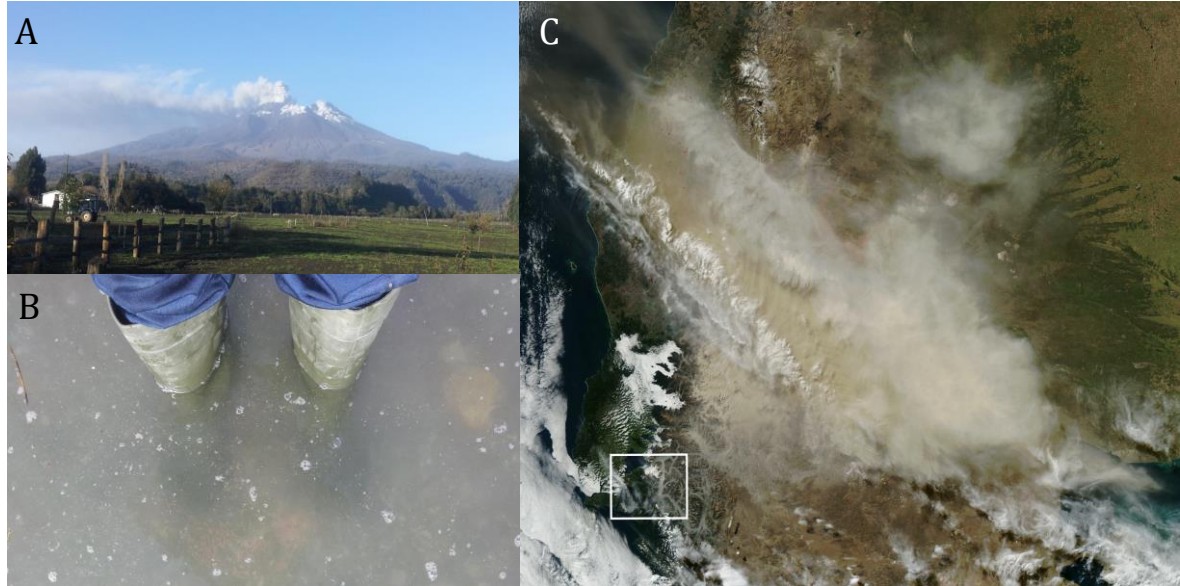

Figure 2. A Calbuco volcano ash plume 6 May 2015. B Reloncaví Fjord water with atypical
high turbidity due to the ash loading, Cochamó town 6 May. C Ash cloud visible on MODIS
Aqua satellite from the NASA Earth Observatory, 23 April
(http://earthobservatory.nasa.gov/NaturalHazards/view.php?id=85767&eocn=home&eoci=
nh). The highlighted box in C corresponds to Fig. 1 C.

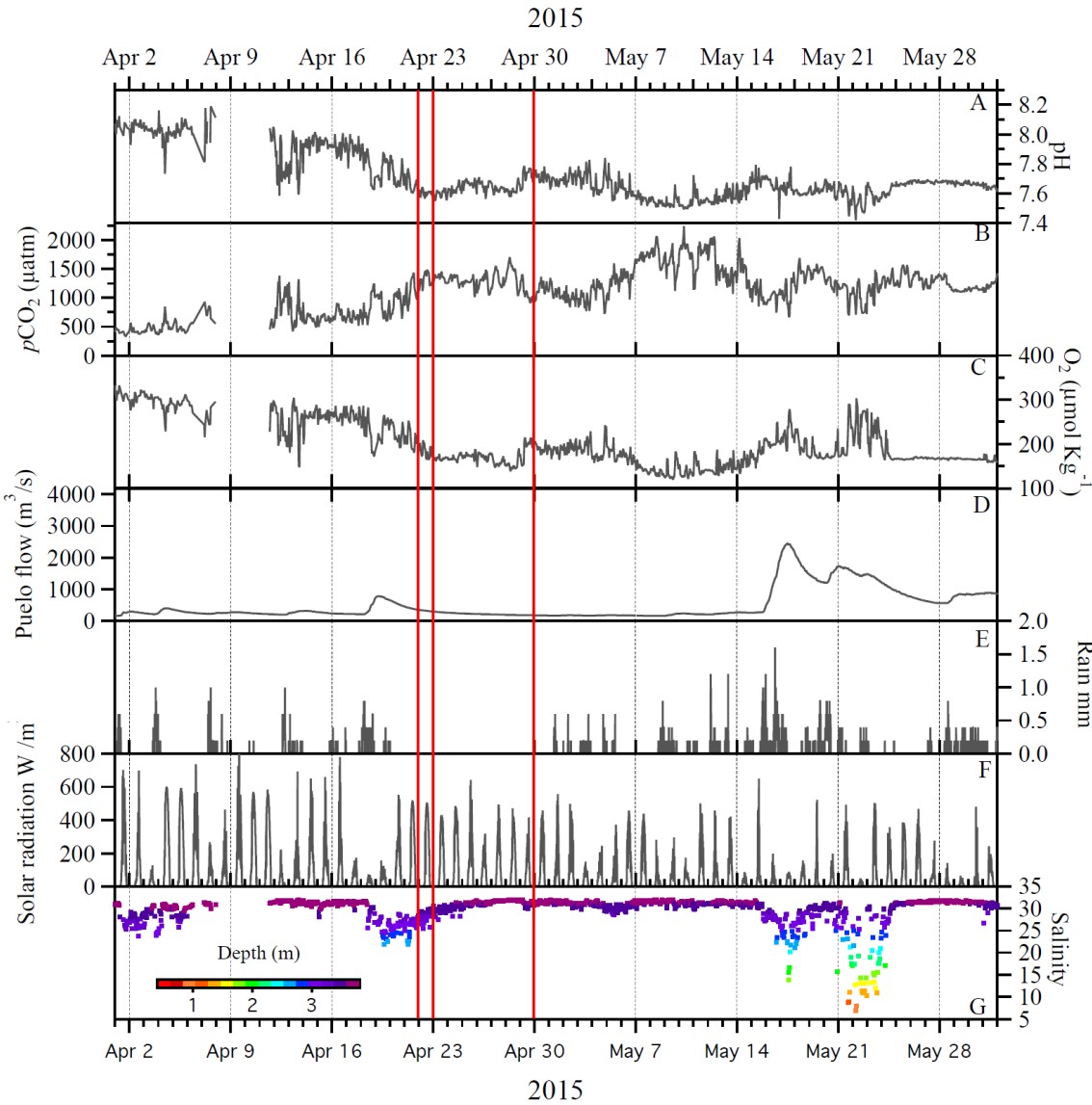


Figure 3. Continuous data from the Reloncaví Fjord mooring and nearby hydrological and
weather stations for April-May 2015. The vertical red lines mark the eruption dates. All
locations are marked in Fig 1. Carbonate chemistry and salinity data from Vergara-Jara et
al., (2019). Wind and tidal mixing caused small changes in the depth of the sensors which
are shown alongside the salinity data.

Carbonate chemistry data from the Reloncaví Fjord mooring demonstrated that pH declined
and $pCO_2$ increased in the week prior to the first eruption (22 April, Fig. 3). Oxygen and pH
reached a minimum and $pCO_2$ a maximum during the time period 7-14 May, which indicates
a state of high respiration. In this stratified environment, the brackish fjord surface layer has
generally low pH, high $pCO_2$ with seasonal changes in salinity and respiration leading to a
large annual range of $pCO_2$ and pH (Vergara-Jara et al., 2019). The depth of the sensors
varied temporally due to changes in tides and river flow. This accounts for some of the
variation in measured salinity due to the strong salinity gradient with depth in the brackish
surface waters (Fig. 3). Any changes to $pCO_2$ or pH occurring as a direct result of the
eruptions, or associated ash deposition, are therefore challenging to distinguish from
background variation due to short-term (intra-day) or seasonal shifts in the carbonate system
which are pronounced in this dynamic and strongly freshwater influenced environment (Fig.
3). Freshwater discharge from the Puelo increased sharply from 16 May which is an annually
recurring event (González et al., 2010).
**3.2 Phytoplankton in Reloncaví fjord post-eruption**

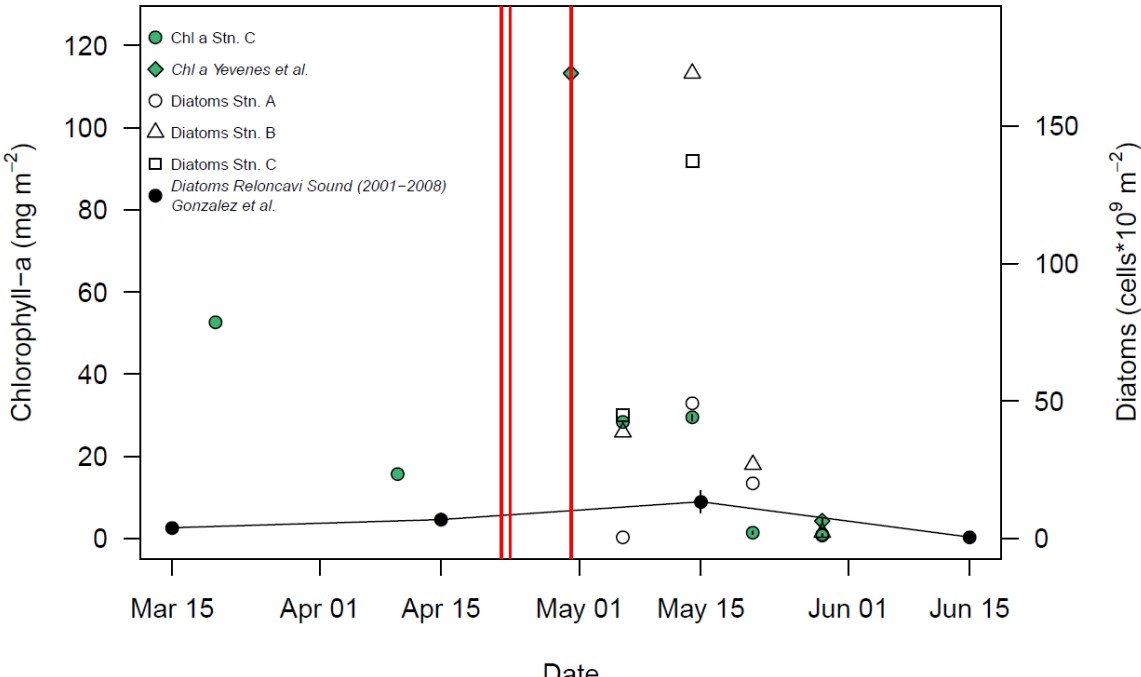


Figure 4. Changes in integrated (0-15 m) diatom abundance and chlorophyll-a for Reloncaví
Fjord in April-May 2015. Locations as per Fig. 1, the eruption dates are marked with red
lines. Historical diatom data from Reloncaví Sound (2001-2008, integrated to 10 m depth,
mean ± standard error, González et al., 2010) and additional chlorophyll data from 2015
('Station 3' from Yevenes et al., 2019, approximately corresponding to Station C herein) are
also shown.

Phytoplankton abundances observed in May 2015 within Reloncaví Fjord were assessed by
diatom cell counts and chlorophyll-a concentrations (Sup. Table 3) and were proportionate
to, or higher than, those previously observed in the region (Fig. 4). When comparing
observations to prior data from González et al., (2010) it should be noted that there is a slight
depth discrepancy (earlier work was integrated to 10 m depth rather than 15 m herein). Yet
as the phytoplankton bloom is overwhelmingly present within the upper 10 m these data do
provide a useful comparison. Diatom abundance integrated to 15 m depth peaked at Stations
B and C around 14 May, with notably lower abundances at the innermost station A (Fig. 4).
The highest measured chlorophyll-a concentrations were on 30 April at Station C, then
chlorophyll-a values declined to much lower concentrations in late May which is expected
from patterns in regional primary production (González et al., 2010). No measurements were
available for 10-30 April 2015 (Fig. 4) and thus it is not possible to determine the timing of
the onset of the austral autumn phytoplankton bloom with respect to the volcanic eruptions
from the available chlorophyll-a or diatom data. Within this time period, the mooring at
Station C (Fig. 3) however did record a modest increase in pH and $O_2$ from 28-29 April,
during a time period when river discharge and salinity were stable, which could be indicative
of the autumn phytoplankton bloom onset.

**3.3 Total alkalinity and macronutrients in leach experiments**
Size analysis of the collected ash determined a mean particle diameter of 339 µm. Small ash-
particles (<63 µm) resulted in minor, or no significant, changes to $A_T$ in brackish fjord waters
(Fig. 5). With larger ash-particles (250-1000 µm) no effect was evident. Conversely, a
leaching experiment with de-ionized water showed a small increase in $A_T$ (Fig. 5) for both
size fractions. By increasing the $A_T$ of freshwater, ash would act to increase the buffering
capacity of river outflow into a typically weak carbonate system like the Reloncaví Fjord
(Vergara-Jara et al., 2019). However, the absolute change in $A_T$ was relatively small despite
the large ash loading used in all incubations ($< 20\,\mu mol\,kg^{-1}$ $A_T$ for ash loading $>4\,g\,L^{-1}$) and
therefore it is expected that the direct effect of ash on $A_T$ in situ was limited. Other effects on
carbonate chemistry may however arise due to ash moderating the timing and intensity of
primary production and thus biological $pCO_2$ drawdown.

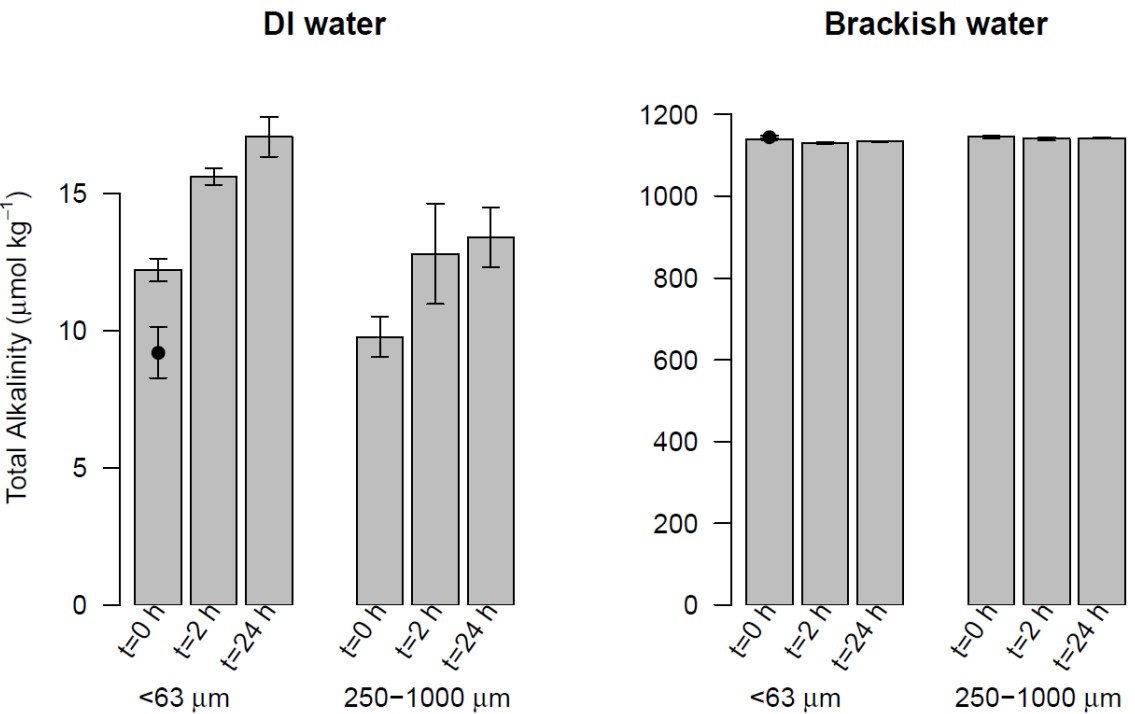


Figure 5. Total alkalinity released after leaching 4.5 g L$^{-1}$ ash of two size fractions (<63 µm
and 250-1000 µm) in de-ionized water (DI water) and brackish water. $T_0$= 'time zero',
measured after one minute of mixing, $T_{2H}$= after two hours of mixing, $T_{24H}$= after 24 hours
of mixing. n=4 for all treatments (mean ± standard deviation plotted). The initial (pre-ash
addition) alkalinity is marked by a black dot superimposed on the left $T_0$. Source data is
provided in Supplementary Table 4.

Ion chromatography results for $Na^+$, $K^+$, $Ca^{2+}$, $F^-$, $Cl^-$, $NO_3^-$ and $SO_4^{2-}$ showed that in the
presence of smaller ash size particles, ion inputs were generally higher (Table 2) as has been
reported previously (Jones and Gislason, 2008; Óskarsson, 1980; Rubin et al., 1994). The
leaching from ash components into de-ionized water occurred almost instantly with limited,
or no increases in leached concentrations observed between 0, 2 and 24 h (Table 2). For larger
particles there was less release of most ions. In the case of $Ca^{2+}$ and $SO_4^{2-}$ a more gradual
leaching effect was apparent (Table 2). The concentrations of $NO_3^-$ and $NH_4^+$ were generally
below detection suggesting that ash was a minor source of fixed-nitrogen into solution. These
observations are consistent with the trends in prior work using a range of volcanic ash and
incubation conditions (Duggen et al., 2010; Witham et al., 2005). Major ion analysis was
only conducted in de-ionized water as no significant changes would be observable for most
of these ions in brackish or saline waters under the same conditions.

Table 2. Major ion and macronutrient concentrations in μmol/l leached from the two size
fractions of ash (< 63 μm and 250-1000 μm) into deionized water (b.d. = below detection).
Shown are mean values, with the standard deviation in parentheses (n=4). Also shown are
mass normalized values [μmol/g ash], and a comparison to the range of values reported by
Jones and Gislason (2008).

| | Time [h] | Na$^+$ | K$^+$ | Ca$^{2+}$ | F$^-$ | Cl$^-$ | SO$_4^{2-}$ | NO$_3^-$ | NH$_4^+$ |
|---|---|---|---|---|---|---|---|---|---|
| *Detection limit* | | *0.17* | *0.43* | *0.30* | *0.28* | *1.31* | *1.64* | *0.34* | *0.13* |
| *Proced. Blank* | | b.d. | b.d. | 0.39 | b.d. | b.d. | b.d. | b.d. | b.d. |
| 250-1000 μm | 0.1 | 3.4 (2.8) | 0.83 (0.3) | 18.3 (3.3) | 0.16 (0.05) | 3.7 (1.9) | 3.7 (2.2) | b.d. | 0.15 (0.2) |
| [μmol/l] | 2 | 5.1 (2.0) | 1.0 (0.2) | 18.5 (4.5) | 0.21 (0.08) | 4.4 (1.6) | 4.9 (2.0) | b.d. | 0.38 (0.4) |
| | 24 | 7.3 (0.1) | 1.4 (0.2) | 23.4 (3.2) | 0.52 (0.18) | 5.7 (0.5) | 8.3 (2.1) | b.d. | b.d. |
| <63 μm | 0.1 | 16.2 (12.7) | 3.2 (0.3) | 25.1 (5.4) | 0.29 (0.0) | 17.1 (13.6) | 13.5 (1.3) | 0.53 (0.2) | 1.70 (1.1) |
| [μmol/l] | 2 | 16.7 (1.0) | 3.8 (0.1) | 31.8 (2.7) | 0.63 (0.2) | 15.2 (0.9) | 19.0 (0.3) | b.d. | 0.52 (1.0) |
| | 24 | 17.3 (0.8) | 3.9 (0.3) | 33.8 (3.3) | 0.69 (0.3) | 14.6 (1.0) | 18.8 (0.5) | b.d. | 1.32 (2.6) |
| <63 μm | 24 | 3.84 | 0.87 | 7.50 | 0.15 | 3.25 | 4.18 | 0.048 | 0.29 |
| [μmol/g ash] | Range (lit.) | 1.5-84.3 | 0.1-5.4 | 0.6-589 | 0.1-9 | 2-92.9 | 1-554 | 0-6.4 | 0.3-0.6 |

**3.4 Trace elements in leach experiments**
Release of nanomolar concentrations of dissolved Fe and Mn was evident when ash was re-
suspended in aged seawater for 10 minutes (Fig. 6). The net release of dissolved metals
proceeded with varying relationships with ash loading over the applied gradient (2-50 mg L$^-$
$^1$). Dissolved Mn, Pb, Cu and Co release exhibited significant (p < 0.05) positive relationships
with ash loading, with Mn and Cu exhibiting the most linear behavior ($R^2$ 0.99 and 0.83,
respectively). Dissolved Fe, Cd and Ni showed no significant relationships with ash loading
over the applied range. The initial concentration of metals in South Atlantic seawater should
however also be considered when interpreting the trends.  The magnitude of changes in Cd
and Ni concentrations were smallest relative to both the initial concentration and the standard
deviation on the initial concentration (0.38 ± 0.04 nM Cd and 6.58 ± 0.76 nM Ni,
respectively). It thus would be difficult to extract a clear relationship irrespective of their
chemical behavior. For other elements (Fe, Co and Pb), non-linearity between ash addition
and trace metal concentrations, and some negative changes in concentrations, both likely
reflect scavenging of metal ions onto ash particle surfaces (Rogan et al., 2016). Fe, Co and
Pb are all scavenged type elements and so increasing the surface area of ash present may
affect the net change in metal concentration. The divergence between the behaviour of Mn
and Fe, with Mn showing a stronger relationship with ash loading, supports the hypothesis
of Mendez et al., (2010), that the release of dissolved Mn from aerosols into seawater depends
primarily on ash Mn availability whereas the release of dissolved Fe is more dependent on
the nature of organic material present in solution.

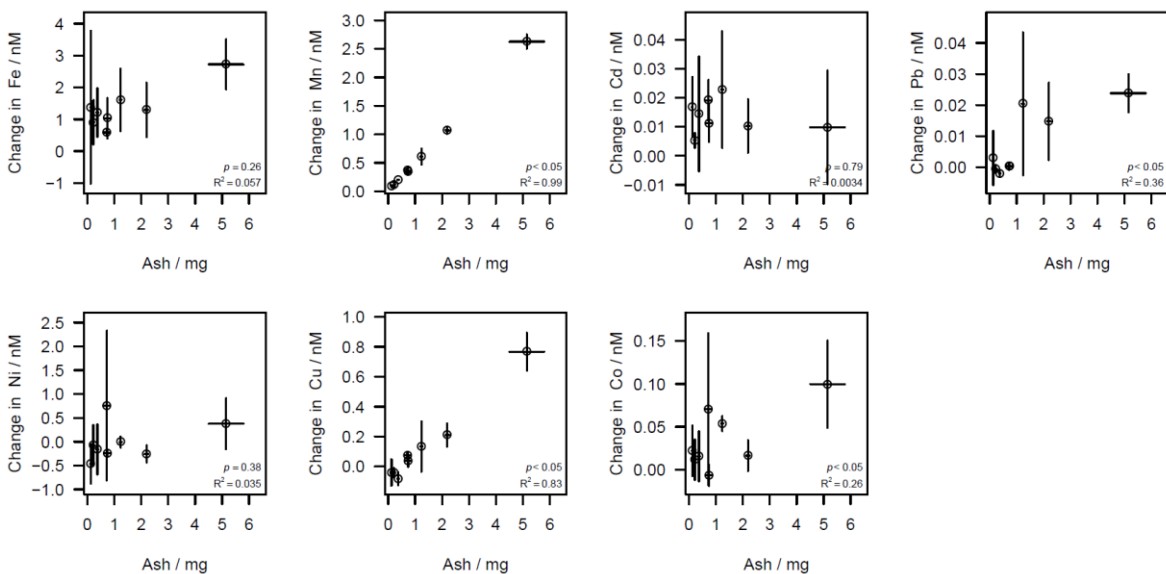


Figure 6. Change in trace metal concentrations after varying ash addition to 100 ml South
Atlantic seawater for a 10-minute leach duration at room temperature. Initial (mean $\pm$
standard deviation) dissolved trace metal concentrations - deducted from the final
concentrations to calculate the change as a result of ash addition - were $0.98 \pm 0.03$ nM Fe,
$0.38 \pm 0.04$ nM Cd, $13 \pm 2$ pM Pb, $6.58 \pm 0.76$ nM Ni, $0.84 \pm 0.07$ nM Cu, $145 \pm 9$ pM Co,
$0.72 \pm 0.05$ nM Mn. Error bars are standard deviations from triplicate treatments with similar
ash loadings. p values and $R^2$ for a linear regression are annotated. Source data is provided
in Supplementary Table 5. The same data with individual replicates is shown in
Supplementary Figure 1.

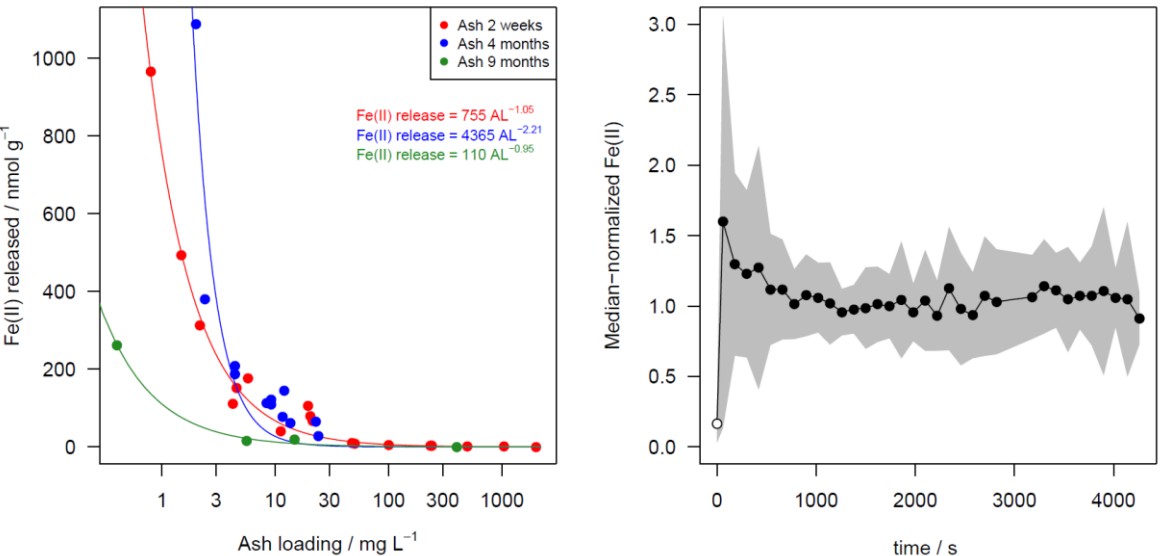


Figure 7. Fe(II) release from Calbuco ash into seawater. Mean Fe(II) released into South
Atlantic seawater over a 30 minute leach at 5-7°C (left). The same batch of Calbuco ash was
subsampled and used to conduct experiments on 3 occasions after the 2015 eruption (2 weeks,
4 months and 9 months since ash collection). The lines are power law fits, with associated
equations shown in the legend. The 3 time-series of Fe(II) concentrations following ash
addition is considered collectively by normalizing the measured concentrations (right), such
that 1.0 represents the median Fe(II) concentration measured in each experiment. All
experiments were conducted for at least 30 minutes, those conducted with 4/9 months old
ash were extended for 1 hour. The black line shows the mean response over 34 leach
experiments with varying ash loading, the shaded area shows ± 1 standard deviation. The
initial Fe(II) concentration (pre-ash addition at 0 s) in all cases was below detection and thus
the detection limit is plotted at 0 s (open circle). Source data is provided in Supplementary
Table 2.

In addition to the release of dFe in solution, which generally exists as Fe(III) species in oxic
seawater (Gledhill and Buck, 2012), the release of Fe(II) was evident on a similar timescale
when cold (5-7°C) aged S Atlantic seawater was used as leachate (Fig. 7). The half-life of
Fe(II) decreases more than tenfold as temperature is increased from 5 to 25°C, leading to
Fe(II) decay on timescales shorter than the time required for analysis (approximately 60 s for
solution to enter the flow injection apparatus, mix with reagent and generate a peak)
(Santana-Casiano et al., 2005). Elevated Fe(II) concentrations (mean 0.8 nM, Sup. Table 2)
were evident at this temperature (5-7°C), which represents an intermediate sea surface
temperature for the high latitude ocean. A sharp decline in Fe(II) dissolution efficiency with
increasing ash load was also evident (Fig. 7). Both the highest Fe(II) concentration and the
highest net release of Fe(II) were observed at the lowest ash loading (Fig. 7 and Sup. Fig. 2).
Fe(II) concentration following dust addition into seawater was possibly reduced when the
same experimental leaches with ash were repeated 9 months after the initial experiment. The
first leaches were conducted ~2 weeks after ash collection. The absence of a clear change
between 2 weeks and 4 months precludes an accurate assessment of the rate at which Fe(II)
solubility may have decreased.

As Fe(II) concentrations were measured continuously using flow injection analysis, the
temporal development of Fe(II) concentration after ash addition to cold seawater can also be
shown (Fig. 7). Considering the set of leach experiments collectively, all ash additions were
characterized by a sharp increase in Fe(II) concentrations in the first minute after ash addition
into seawater. This was typically followed by a decline and then a relatively stable Fe(II)
concentration (Fig. 7).
**3.5 Satellite observations**
Five-day composite images of atmospheric aerosol loading (UV aerosol index, which largely
represents strongly UV-absorbing dust, Torres et al., 2007) indicated two main volcanic
eruption plume trajectories following the major eruptions on 22 and 23 April: (i) northwards
over the Pacific, and (ii) northeast over the Atlantic. Daily resolved time series were
constructed for regions in the Atlantic and Pacific with elevated atmospheric aerosol loading
(UV Aerosol Index ~2 a.u.; Fig. 8). The Pacific time series indicated a pronounced peak in
aerosol index followed by chlorophyll-a one day later. A control region to the south of the
ash-impacted Pacific region showed no clear changes in chlorophyll-a matching that
observed in the higher UV aerosol index region to the north (Sup. Fig. 3).

Conversely, in the Atlantic, where the background chlorophyll-a concentration was higher
throughout the time period of interest, the main area with enhanced aerosol index was not
clearly associated with a change in chlorophyll-a dynamics on a timescale comparable to that
observed following other volcanic ash fertilized events (Fig. 8). In a smaller ash impacted
area to the south of the Rio de la Plata (Sup. Fig. 3), where nitrate levels are expected to be
higher than to the north and Fe levels also expected to be elevated due its location on the
continental shelf, a chlorophyll-a peak was evident 7 days after the UV aerosol peak.
However, this was not well constrained due to poor satellite coverage in the period after the
eruption.

Prior eruptions have been attributed with driving time periods of enhanced regional marine
primary production beginning 3-5 days post-eruption (Hamme et al., 2010; Langmann et al.,
2010; Lin et al., 2011) and bottle experiments showing positive chlorophyll changes in
response to ash addition are typically significant compared to controls within 1-4 days
following ash addition (Browning et al., 2014; Duggen et al., 2007; Mélançon et al., 2014).

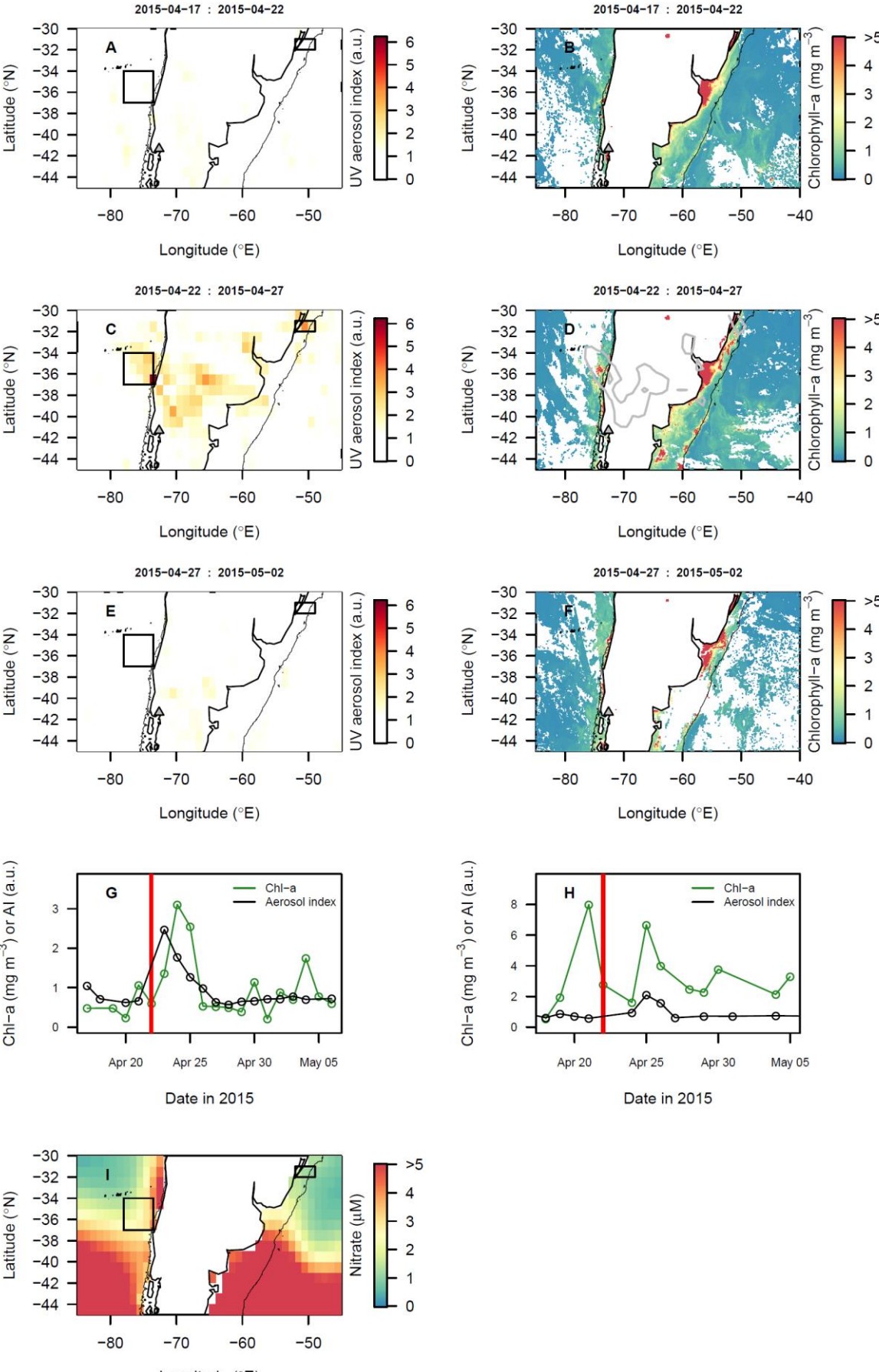


Figure 8. Potential biological impact of the 2015 Calbuco eruption observed via satellite
remote sensing. (A-F) Spatial maps showing the distribution of ash in the atmosphere (UV
Aerosol Index) and corresponding images of chlorophyll-a. Images were composited over 5-
day periods. Grey lines in chlorophyll maps corresponds to the UV Aerosol index = 2 a.u.
contour. (G, H) Time series of UV Aerosol Index and chlorophyll-a for regions of the Pacific
(G) and Atlantic (H) identified by boxes in maps. Red vertical lines (22 April) indicate the
first eruption date. (I) Mean World Ocean Atlas surface $NO_3$ concentrations. Thin black lines
indicate the 500 m bathymetric depth contour.

**4 Discussion**
**4.1 Local drivers of 2015 bloom dynamics in Reloncaví Fjord**
The north Patagonian archipelago and fjord region have a seasonal phytoplankton bloom
cycle with peaks in productivity occurring in May and October (austral autumn and spring)
and the lowest productivity consistently in June (austral winter)(González et al., 2010).
Diatoms normally dominate the phytoplankton community during the productive period due
to high light availability and high silicic acid supply, both of which are influenced by
freshwater runoff (González et al., 2010; Torres et al., 2014). The austral fall season,
encompassing the April-May 2015 ash deposition events, is therefore expected to have a high
phytoplankton biomass (Iriarte et al., 2007; León-Muñoz et al., 2018) which terminates
abruptly with decreasing light availability in austral winter (González et al., 2010).

Whilst not directly comparable, the magnitude of the 2015 bloom in terms of diatom
abundance (Fig. 4) was more intense than that reported in Reloncaví Sound 2001-2008. With
respect to the timing of the phytoplankton bloom, the low diatom abundances and
chlorophyll-a concentrations at the end of May (Fig. 4) are consistent with prior observations
of sharp declines in primary production moving into June (González et al., 2010). Peaks in
diatom abundance were measured at two stations on 14 May one week after the third (small)
eruptive pulse, and measured chlorophyll-a concentrations were highest close to Station C
on 30 April (Fig. 4). The high-resolution pH and $O_2$ data collected at Station C from mooring
data is consistent with an intense phytoplankton bloom between ~29 April and 7 May (Fig.
3) indicated by a shift to slightly higher pH and $O_2$ during this time period when river flow
into the fjord was stable.

Without a direct measure of ash deposition per unit area in the fjord, turbidity, or higher
resolution chlorophyll/diatom data, it is challenging to unambiguously determine the extent
to which the austral autumn phytoplankton bloom was affected by volcanic activity. The high
abundance of diatoms at two of three stations sampled could have resulted from ash
fertilization. Yet if this was the case, it is not clear which nutrient was responsible for this
fertilization, why the bloom initiation occurred about one week after the third eruptive pulse
(several weeks after the main eruption events) and to what extent the timing was coincidental
given that productivity normally peaks in May. Reloncaví Fjord was to the south of the
dominant ash deposition from the 22 and 23 April eruptions (Romero et al., 2016) and thus
ash was delivered by a mixture of vectors including runoff and rainfall. The Petrohue river
basin was particularly severely affected by ash with deposition of up to 50 cm ash in places.
This complicates the interpretation of the time series provided by high resolution data (Fig.
3). With incident light also highly variable over the time series (Fig. 3F), there are clearly
several factors, other than volcanic ash deposition, which will have exerted some influence
on diatom and chlorophyll-a abundance throughout May 2015.

Primary production in the Reloncaví region is thought to be limited by light availability rather than macronutrient availability (González et al., 2010). Whilst micronutrient availability relative to phytoplankton demand has not been extensively assessed in this fjord, with such higher riverine inputs across the region- which are normally a large source of dissolved trace elements into coastal waters (e.g. Boyle et al., 1977)- limitation of phytoplankton growth by Fe, or another micronutrient, seems implausible. Reported Fe concentrations determined by a diffusive gel technique in Reloncaví Fjord in October 2006 were relatively high; 46-530 nM (Ahumada et al., 2011). Similarly, reported dFe concentrations in the adjacent Comau Fjord at higher salinity are generally in the nanomolar range and remain >2 nM even under post-bloom conditions which suggests dFe is not a limiting factor for phytoplankton growth (Hopwood et al., 2020; Sanchez et al., 2019).

Silicic acid availability could have been increased by ash deposition. Whilst not quantified herein, an increase in silicic acid availability from ash in a region where silicic acid was sub-optimal for diatom growth could plausibly explain higher than usual diatom abundance (Siringan et al., 2018). Silicic acid concentrations were indeed high (up to 118 µM) in Reloncaví Fjord surface waters and >30 µM at 15 m depth (salinity 33.4) (Vergara-Jara et al., 2019; Yevenes et al., 2019). However concentrations >30 µM are typical during periods of high runoff and accordingly are not thought to limit primary production or diatom growth in this area (González et al., 2010). The $Si(OH)_4$:$NO_3$ ratio in Reloncaví Fjord and downstream Reloncaví Sound also indicates an excess of $Si(OH)_4$, with ratios of approximately 2:1 observed in fjord surface waters throughout the year (González et al., 2010; Yevenes et al., 2019). For comparison, the ratio of Si:N for diatom nutrient uptake is

15:16 (Brzezinski, 1985). Furthermore, experimental incubations making additions of
macronutrients to fjord waters in Reloncaví and adjacent fjords, have found strong responses
of phytoplankton to additions of silicic acid only when $Si(OH)_4$ and $NO_3$ were added in
combination, further corroborating the hypothesis that an excess of silicic acid is normally
present in surface waters of these fjord systems (Labbé-Ibáñez et al., 2015). It is therefore
doubtful that changes in nutrient availability from ash alone could explain such high diatom
abundances in mid-May.

Alternative reasons for high diatom abundances in the absence of a chemical fertilization
effect are plausible and could include, for example, ash having reduced zooplankton
abundance or virus activity in the fjord, thus facilitating higher diatom abundance than would
otherwise have been observed by decreasing diatom mortality rates in an environment where
nutrients were replete. The role of volcanic ash in driving such short-term ecological shifts
in the marine environment is almost entirely unstudied (Weinbauer et al., 2017). However,
volcanic ash deposition of 7 mg $L^{-1}$ in lakes within this region during the 2011 Puyehue-
Cordón Caulle eruption was reported to increase post-deposition phytoplankton biomass and
decrease copepod and cladoceran biomass (Wolinski et al., 2013). The proposed mechanism
was ash particle ingestion negatively affecting zooplankton, and ash-shading positively
affecting phytoplankton via reduced photoinhibition (Balseiro et al., 2014; Wolinski et al.,

633 2013).


Considering the more modest peak in diatom abundance at the most strongly ash affected
station (Station A, Fig. 4) and the timing of the peak diatom abundance 3 weeks after the
main eruption, it is clear that the interaction between ash and phytoplankton in the Reloncaví
Fjord was more complex than the simple Fe-fertilization proposed for the SE Pacific (Fig.
8g). In the absence of an immediate diatom fertilization effect from Fe or silicic acid, we
hypothesize that any change in phytoplankton bloom dynamics within Reloncaví Fjord was
mainly a 'top-down' effect driven by the physical interaction of ash and different ecological
groups in a nutrient replete environment, rather than a 'bottom-up' effect driven by
alleviation of nutrient-limitation from ash dissolution.
**4.2 Volcanic ash as a unique source of trace elements**
The release of the bioessential elements Fe and Mn from ash here ranged from 53 - 1200
nmol $g^{-1}$ (dFe) and 48 - 71 nmol $g^{-1}$ (dissolved Mn). For dFe this is comparable to the rates
determined in other studies under similar experimental conditions for subduction zone
volcanic ash, with reported Fe-release in prior work ranging 2-570 nmol $g^{-1}$ (Sup. Table 1).
For Mn, less prior work is available, but these values are within the 17-1300 nmol $g^{-1}$ range
reported by Hoffmann et al., (2012). Fe(II) release was particularly efficient at ash loadings
<5 mg $L^{-1}$ (Fig. 7), whereas dFe release was less sensitive to ash loading (Fig. 6). The timing
of Fe(II) release in the first 60 s of incubations suggests a fast dissolution process. Fe(II) is
short lived in oxic surface seawater with an observed half-life of only 10-20 minutes even in
the Southern Ocean where cold surface waters slow Fe(II) oxidation (Sarthou et al., 2011).
Yet, relative to Fe(III), Fe(II) is also more soluble and, from an energetic perspective,
expected to be more bioaccessible to cellular uptake (Sunda et al., 2001). Whilst it is known
that the vast majority of dFe leached from ash into seawater tends to occur in the first minutes
of ash addition (Duggen et al., 2007; Jones and Gislason, 2008), and this could be consistent
with rapid dissolution of highly soluble phases on ash surfaces, we note that there is not yet
conclusive evidence concerning the precise origin of this dFe pulse. Fe(II) salts may be
present on the surface of ash particles (Horwell et al., 2003; Hoshyaripour et al., 2015) and
thus the Fe(II) observed herein (Fig. 7) may reflect almost instantaneous release following
dissolution of thin layers of salt coatings in ash surfaces (Ayris and Delmelle, 2012; Delmelle
et al., 2007; Olsson et al., 2013). Alternatively Fe(II) could be released from more crystalline
Fe(II) phases. Prior work, at much lower pH (pH 1 $H_2SO_4$ representing conditions that ash
surfaces may experience during atmospheric processing, but not in aquatic environments)
suggests that short-term release of Fe(II) or Fe(III) is determined by the surface Fe(II)/Fe
ratio which may differ from the bulk Fe(II)/Fe ratio due to plume processing (Maters et al.,

669    2017).

Different leaching protocols are widely recognised as a major challenge for interpreting and
comparing different dissolution experiment datasets for all types of aerosols (Duggen et al.,
2007; Morton et al., 2013). When Fe(II) is released into solution as a considerable fraction
of the total dFe release this is particularly challenging to monitor, as Fe(II) oxidises on
timescales of seconds to minutes depending on temperature, pH and $O_2$ conditions (Santana-
Casiano et al., 2005). The dFe and Fe(II) leaching protocols used herein are only comparable
qualitatively, as the Fe(II) method using cooler seawater and larger seawater volumes was
specifically designed to test for the presence of rapid Fe(II) release and to evaluate the short-
term temporal trend of any such release. Yet, for rough comparative purposes, the Fe(II)
released was equivalent to $38 \pm 25\%$ (mean $\pm$ standard deviation) of dFe released at ash
loadings from 1-10 mg $L^{-1}$ and $19 \pm 17\%$ of dFe for ash loadings from 10-50 mg $L^{-1}$. These
values are reasonably comparable to the 26% median Fe(II)/dFe fraction measured in Fe
released into seawater from aerosols collected across zonal transects of the Pacific Ocean
(Buck et al., 2013) suggesting that fresh Calbuco ash is roughly comparable in terms of Fe(II)
lability to these environmentally processed aerosols.
**4.3 A potential fertilization effect in the SE Pacific**
Experiments with ash suspensions have shown that ash loading has a restricted impact on
satellite chlorophyll-a retrieval (Browning et al., 2015), therefore offering a means to assess
the potential biological impact of the 2015 Calbuco eruption in offshore waters. We found
evidence for fertilization of offshore Pacific seawaters in the studied area (Fig. 8). Following
the eruption date, mean chlorophyll-a concentrations increased ~2.5 times over a broad
region where elevated UV aerosol index was detected (Fig. 8G). Both the timing and location
of this chlorophyll-a peak were consistent with ash fertilization, with the peak of elevated
chlorophyll-a being located within the core of highest atmospheric aerosol loading, and the
peak date occurring one day after the main passage of the atmospheric aerosol plume. A
similar phytoplankton response timeframe was reported following ash deposition in the NE
Pacific following the August 2008 Kasatochi eruption (Hamme et al., 2010) which was
similarly thought to be triggered by relief of Fe-limitation (Langmann et al., 2010). At the
same time, a control region to the south of the ash-impacted Pacific region showed no clear
changes in chlorophyll-a matching that observed in the higher UV aerosol index region to
the north (Sup. Fig. 3).

In the SW Atlantic, two ash impacted areas are highlighted; one to the north (Fig. 8), and one
to the south of the Rio de la Plata (Sup. Fig. 3). Nitrate levels are expected to be higher in the
south than to the north, with Fe levels expected to be elevated across both locations as a result
of their position on the continental shelf. In the area to the north of the Rio de la Plata (Fig.
8), ash deposition indicated by the UV aerosol index did not lead to such a clear
corresponding change in chlorophyll-a concentrations (Fig. 8H), although with the available
data it is not possible to rule out the possibility of fertilisation completely (e.g., whilst also
being proceeded by a larger chlorophyll-a peak on 21 April, there is a peak in chlorophyll-a
at 25 April coinciding with elevated UV aerosol index). Phytoplankton growth in this region
of the Atlantic is expected to be limited by fixed nitrogen availability, as a result of strong
stratification (Moore et al., 2013) and thus dFe release from ash particles alone would not be
expected to result in short-term increases to primary production. In the second area of ash
deposition, to the south (Sup. Fig. 3), a chlorophyll-a peak was evident 7 days after the UV
aerosol peak. However, this was not well constrained due to poor satellite coverage in the
period after the eruption. Considering the dynamic spatial and temporal variation in
chlorophyll within this coastal area, it is challenging to associate any change in chlorophyll
specifically with ash arrival.

The change in chlorophyll-a observed in the SE Pacific contrasts with results in Reloncaví
Fjord where phytoplankton abundances likely peaked much later than the first ash arrival-
after 28 April. The fertilized region of the Pacific (Fig. 8) hosts upwelling of deep waters,
supplying nutrients in ratios that are deficient in dFe (Bonnet et al., 2008; Torres and
Ampuero, 2009). Fe-limitation of phytoplankton growth in this region is therefore
anticipated, which could have been temporarily relieved following ash deposition and dFe
release (Fig. 6). The differential responses observed in the Pacific and Atlantic are therefore
consistent with the anticipated nutrient limitation regimes (Fe-limited and nitrogen-limited,
respectively), and the supply of dFe but not fixed N ($NO_3$ or $NH_4$) from the Calbuco ash (Fig.
6 and Table 2).

**5 Conclusions**
The contrasting effects of volcanic ash on primary producers in Reloncaví Fjord, the SE
Pacific and SW Atlantic Oceans support the hypothesis that the response of primary
producers is dependent on both the ash loading and the resources limiting primary production
in a region at a specific time of year. Leach experiments using ash from the 2015 Calbuco
eruption demonstrated a small increase in the alkalinity of de-ionized water from fine, but
not coarse ash, and no significant addition of fixed nitrogen (quantified as $NO_3$ and $NH_4$) into
solution. In saline waters, release of dissolved trace metals including Mn, Cu, Co, Pb, Fe and
specifically Fe(II) was evident.

Strong evidence of a broad-scale 'bottom-up' fertilization effect of ash on phytoplankton was
not found locally within Reloncaví Fjord, although it is possible that the timing and peak
diatom abundance of the autumn phytoplankton bloom may have shifted in response to high
ash loading in the weeks following the first eruption. High diatom abundances at some
stations within the fjord several weeks after the eruption may have arisen from a 'top-down'
effect of ash on filter feeders, although the mechanism can only be speculated herein. No
clear positive effect of ash deposition on chlorophyll-a was evident in the SW Atlantic,
consistent with expected patterns in nutrient deficiency which suggest the region to be
nitrogen-limited. However, in offshore waters of the SE Pacific where Fe is anticipated to
limit phytoplankton growth, a chlorophyll-a increase was related with maximum ash
deposition and we presume that this increase in chlorophyll-a was likely driven by Fe-
fertilization.

**6. Data availability**
The complete 2015 time series from the Reloncaví Fjord mooring is available online
(https://figshare.com/articles/Puelo_Bouy/7754258). Source data for Figures 4-7 is included
in the Supplement.

**7. Acknowledgements**

The authors thank the Dirección de Investigación & Desarrollo UACh for its partial support during this project. The data presented are part of the second chapter of the PhD Thesis of MVJ at Universidad Austral de Chile. Cristian Vargas (Universidad de Concepción) is thanked for making additional chlorophyll a data available, Manuel Díaz for providing Fig. 1, Lorena Rebolledo for running the particle size test, Miriam Beck for assistance with Fe(II) flow injection analysis and 3 reviewers for constructive comments that improved the manuscript.

**8. Funding**

JLI and EA gratefully acknowledge funding from the European Commission (OCEAN-CERTAIN, FP7- ENV- 2013-6.1-1; no: 603773). JLI received funding by CONICYT-FONDECYT 1141065 and is partially funded by Center IDEAL (FONDAP 15150003). Partial funding came from CONICYT-FONDECYT 1140385 (RT). MVJ received financial support from a CONICYT Scholarship (Beca Doctorado Nacional 2015 No 21150285). IR and MH received funding from the Deutsche Forschungsgemeinschaft as part of Sonderforschungsbereich (SFB) 754: 'Climate-Biogeochemistry Interactions in the Tropical Ocean'.

**9. Author contributions**

MVJ, MH, JLI and EA designed the study. MVJ, IR, MH, RT and BR conducted analytical and field work. TB conducted satellite data analysis. MV, MH and TB wrote the initial manuscript with all authors contributing to its revision.

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

Methods for the sampling and analysis of marine aerosols: Results from the 2008
GEOTRACES aerosol intercalibration experiment, Limnol. Oceanogr. Methods, 11,
doi:10.4319/lom.2013.11.62, 2013.
Mosley, L. M., Husheer, S. L. G. and Hunter, K. A.: Spectrophotometric pH measurement
in estuaries using thymol blue and m-cresol purple, Mar. Chem., 91, 175–186,
doi:10.1016/j.marchem.2004.06.008, 2004.
Newcomb, T. W. and Flagg, T. A.: Some effects of Mt. St. Helens volcanic ash on juvenile
salmon smolts., Mar. Fish. Rev., 45(2), 8–12, 1983.
Olgun, N., Duggen, S., Croot, P. L., Delmelle, P., Dietze, H., Schacht, U., Óskarsson, N.,
Siebe, C., Auer, A. and Garbe-Schönberg, D.: Surface ocean iron fertilization: The role of
airborne volcanic ash from subduction zone and hot spot volcanoes and related iron fluxes
into the Pacific Ocean, Global Biogeochem. Cycles, 25(4), doi:10.1029/2009GB003761,

956 2011.

Olsson, J., Stipp, S. L. S., Dalby, K. N. and Gislason, S. R.: Rapid release of metal salts and
nutrients from the 2011 Grímsvötn, Iceland volcanic ash, Geochim. Cosmochim. Acta,
doi:10.1016/j.gca.2013.09.009, 2013.
Óskarsson, N.: The interaction between volcanic gases and tephra: Fluorine adhering to
tephra of the 1970 hekla eruption, J. Volcanol. Geotherm. Res., doi:10.1016/0377-

962 0273(80)90107-9, 1980.

Rapp, I., Schlosser, C., Rusiecka, D., Gledhill, M. and Achterberg, E. P.: Automated
preconcentration of Fe, Zn, Cu, Ni, Cd, Pb, Co, and Mn in seawater with analysis using
high-resolution sector field inductively-coupled plasma mass spectrometry, Anal. Chim.
Acta, 976, 1–13, doi:10.1016/j.aca.2017.05.008, 2017.
Reckziegel, F., Bustos, E., Mingari, L., Báez, W., Villarosa, G., Folch, A., Collini, E.,
Viramonte, J., Romero, J. and Osores, S.: Forecasting volcanic ash dispersal and coeval
resuspension during the April-May 2015 Calbuco eruption, J. Volcanol. Geotherm. Res.,
doi:10.1016/j.jvolgeores.2016.04.033, 2016.
Rogan, N., Achterberg, E. P., Le Moigne, F. A. C., Marsay, C. M., Tagliabue, A. and
Williams, R. G.: Volcanic ash as an oceanic iron source and sink, Geophys. Res. Lett.,
43(6), 2732–2740, doi:10.1002/2016GL067905, 2016.
Romero, J. E., Morgavi, D., Arzilli, F., Daga, R., Caselli, A., Reckziegel, F., Viramonte, J.,
Díaz-Alvarado, J., Polacci, M., Burton, M. and Perugini, D.: Eruption dynamics of the 22–
23 April 2015 Calbuco Volcano (Southern Chile): Analyses of tephra fall deposits, J.
Volcanol. Geotherm. Res., 317, 15–29, doi:10.1016/j.jvolgeores.2016.02.027, 2016.
Rubin, C. H., Noji, E. K., Seligman, P. J., Holtz, J. L., Grande, J. and Vittani, F.:
Evaluating a fluorosis hazard after a volcanic eruption, Arch. Environ. Health,
doi:10.1080/00039896.1994.9954992, 1994.
Sanchez, N., Bizsel, N., Iriarte, J. L., Olsen, L. M. and Ardelan, M. V.: Iron cycling in a
mesocosm experiment in a north Patagonian fjord: Potential effect of ammonium addition
by salmon aquaculture, Estuar. Coast. Shelf Sci., 220, 209–219,
doi:10.1016/j.ecss.2019.02.044, 2019.
Santana-Casiano, J. M., Gonzaalez-Davila, M. and Millero, F. J.: Oxidation of nanomolar
levels of Fe(II) with oxygen in natural waters, Environ. Sci. Technol., 39(7), 2073–2079,
doi:10.1021/es049748y, 2005.
Sarmiento, J. L.: Atmospheric CO2 stalled, Nature, doi:10.1038/365697a0, 1993.
Sarthou, G., Bucciarelli, E., Chever, F., Hansard, S. P., Gonzalez-Davila, M., Santana-
Casiano, J. M., Planchon, F. and Speich, S.: Labile Fe(II) concentrations in the Atlantic
sector of the Southern Ocean along a transect from the subtropical domain to the Weddell
Sea Gyre, Biogeosciences, 8(9), 2461–2479, doi:10.5194/bg-8-2461-2011, 2011.
Seidel, M. P., DeGrandpre, M. D. and Dickson, A. G.: A sensor for in situ indicator-based
measurements of seawater pH, Mar. Chem., 109(1), 18–28,
doi:10.1016/j.marchem.2007.11.013, 2008.
Simonella, L. E., Palomeque, M. E., Croot, P. L., Stein, A., Kupczewski, M., Rosales, A.,
Montes, M. L., Colombo, F., García, M. G., Villarosa, G. and Gaiero, D. M.: Soluble iron
inputs to the Southern Ocean through recent andesitic to rhyolitic volcanic ash eruptions
from the Patagonian Andes, Global Biogeochem. Cycles, 29(8), 1125–1144,
doi:10.1002/2015GB005177, 2015.
Siringan, F. P., Racasa, E. D. R., David, C. P. C. and Saban, R. C.: Increase in Dissolved
Silica of Rivers Due to a Volcanic Eruption in an Estuarine Bay (Sorsogon Bay,
Philippines), Estuaries and Coasts, 41, 2277–2288, doi:10.1007/s12237-018-0428-1, 2018.
Stewart, C., Johnston, D. M., Leonard, G. S., Horwell, C. J., Thordarson, T. and Cronin, S.
J.: Contamination of water supplies by volcanic ashfall: A literature review and simple
impact modelling, J. Volcanol. Geotherm. Res., 158(3), 296–306,
doi:10.1016/j.jvolgeores.2006.07.002, 2006.
Sunda, W. G., Buffle, J. and Van Leeuwen, H. P.: Bioavailability and Bioaccumulation of
Iron in the Sea, in The Biogeochemistry of Iron in Seawater, vol. 7, edited by D. R. Turner
and K. A. Hunter, pp. 41–84, John Wiley & Sons, Ltd, Chichester., 2001.
Torres, O., Tanskanen, A., Veihelmann, B., Ahn, C., Braak, R., Bhartia, P. K., Veefkind, P.
and Levelt, P.: Aerosols and surface UV products form Ozone Monitoring Instrument
observations: An overview, J. Geophys. Res. Atmos., doi:10.1029/2007JD008809, 2007.
Torres, R. and Ampuero, P.: Strong CO2 outgassing from high nutrient low chlorophyll
coastal waters off central Chile (30°S): The role of dissolved iron, Estuar. Coast. Shelf Sci.,
83(2), 126–132, doi:10.1016/j.ecss.2009.02.030, 2009.
Torres, R., Silva, N., Reid, B. and Frangopulos, M.: Silicic acid enrichment of subantarctic
surface water from continental inputs along the Patagonian archipelago interior sea (41-
56°S), Prog. Oceanogr., 129, 50–61, doi:10.1016/j.pocean.2014.09.008, 2014.
Utermöhl, H.: Zur Vervollkommnung der quantitativen Phytoplankton-Methodik, SIL
Commun. 1953-1996, doi:10.1080/05384680.1958.11904091, 1958.
Vergara-Jara, M. J., DeGrandpre, M. D., Torres, R., Beatty, C. M., Cuevas, L. A., Alarcón,
E. and Iriarte, J. L.: Seasonal Changes in Carbonate Saturation State and Air-Sea CO2
Fluxes During an Annual Cycle in a Stratified-Temperate Fjord (Reloncaví Fjord, Chilean
Patagonia), J. Geophys. Res. Biogeosciences, 124(9), 2851–2865,
doi:10.1029/2019JG005028, 2019.
Watson, A. J.: Volcanic iron, CO2, ocean productivity and climate, Nature,
doi:10.1038/385587b0, 1997.
Weinbauer, M. G., Guinot, B., Migon, C., Malfatti, F. and Mari, X.: Skyfall - neglected
roles of volcano ash and black carbon rich aerosols for microbial plankton in the ocean, J.
Plankton Res., 39(2), 187–198, doi:10.1093/plankt/fbw100, 2017.
Welschmeyer, N. A.: Fluorometric analysis of chlorophyll a in the presence of chlorophyll
b and pheopigments, Limnol. Oceanogr., doi:10.4319/lo.1994.39.8.1985, 1994.
Witham, C. S., Oppenheimer, C. and Horwell, C. J.: Volcanic ash-leachates: a review and
recommendations for sampling methods, J. Volcanol. Geotherm. Res., 141(3), 299–326,
doi:10.1016/j.jvolgeores.2004.11.010, 2005.
Wolinski, L., Laspoumaderes, C., Bastidas Navarro, M., Modenutti, B. and Balseiro, E.:
The susceptibility of cladocerans in North Andean Patagonian lakes to volcanic ashes,
Freshw. Biol., 58, 1878–1888, doi:10.1111/fwb.12176, 2013.
Yevenes, M. A., Lagos, N. A., Farías, L. and Vargas, C. A.: Greenhouse gases, nutrients
and the carbonate system in the Reloncaví Fjord (Northern Chilean Patagonia):
Implications on aquaculture of the mussel, Mytilus chilensis, during an episodic volcanic
eruption, Sci. Total Environ., doi:10.1016/j.scitotenv.2019.03.037, 2019.