# Peer review of "A mosaic of phytoplankton responses across Patagonia, the southeast Pacific and"

_Ocean Science, 2020_

## Referee Comment (RC1) · Anonymous Referee #1 · 3 Aug 2020

"A mosaic of phytoplankton responses across Patagonia, the SE Pacific and SW Atlantic Ocean to ash deposition and trace metal release from the Calbuco 2015 volcanic eruption" by Vergara-Jara et al.

The present study by Vergara-Jara and co-workers is based on collected ash from the 2015 Calbuco volcano emission and water samples from Reloncaví fjord from Patagonia (Chile) and seawater samples from south Atlantic. In this study, the authors did

some leaching experiments using the collected ash and water samples (or deionized water) to understand the significance of fresh and aged ash in its efficiency in leaching trace elements (Fe, Cd, Pb, Ni, Cu, Co, Mn). Besides, to understand the leachability of bioavailable Fe, the Fe(II) fraction, the authors also performed some specific leaching procedures. Also, some additional leaching experiments were performed to understand the changes in major ion abundances and total alkalinity of water samples upon ash addition. Further, in order to understand the changes in primary productivity in the nearby oceanic basin (Atlantic Ocean), satellite data was used.

The present manuscript opens up a study on one of the important aspects in trace metal biogeochemistry: the role/impact of sporadic and random volcanic eruptions and ensuing ash fallout on leaching of trace elements in the surface ocean (and marginal) waters and associated effects on marine biological productivity. The authors have used lab-based leaching experiments to estimate the fraction of dissolved Fe (dFe) and Fe(II) that is susceptible to leaching from the ash. One of the most interesting results of this study is the reduction in release of Fe(II) with aged ash.

The manuscript is overall well written. The statements and arguments are mostly laid out clearly and easy to follow. However, I have some concerns with the present version of the manuscript. My concerns with the present version of the manuscript are two-pronged. These have been detailed as follows: 1) As is in the present version of the manuscript, section 2 (Materials and methods) is difficult to follow. When I started reading this section, I was looking for a common subsection detailing all the samples (and their sampling location) for the present study. Opening section 2, subsection 2.1 is placed well and describes the study area providing the regional details. However, I would like to suggest the authors insert a subsection 2.2, providing details of all the samples collected and analysed in the present work. In addition, there are some minor concerns with this section (section 2). For e.g. (i) this study is based on the changes in biogeochemistry of Reloncaví fjord and the Atlantic Ocean immediately after the eruption of Calbuco volcano. However, some samples from another location

(Aysén fjord) were also discussed in the manuscript (line No. 184). This location has only been mentioned here and nowhere else in the manuscript. (ii) As mentioned in line 134, there is a mooring station located in Reloncaví fjord, its location is not known until late in the manuscript (Line 311). 2) I have some concerns regarding the leaching experiments done by the authors. As is the case in the present work, different leaching experiments were performed for different set of parameters (major ions, trace metals and Fe(II)). Why did the authors use different leaching procedures for different parameters in the present work? It was the same ash falling over the waters of the fjord and Pacific ocean. So, to see the combined effect of the ash falling on fjord waters, authors should have used similar leaching protocols for major ions, trace metals and Fe(II). Also, the authors have used deionized water for their leaching experiments for major ions. Why did the authors not consider using the trace metal free seawater for their leaching experiments? Also, some of the methods are not clear. For the leaching experiments for major ions, the authors have mentioned correcting the abundances for initial water concentrations. Was this also done for trace metal leaching experiments? The authors have mentioned that for leaching experiments for major ions, they used both fjord waters and deionized water, but table 1 only provides results for leaching with deionized water. Also, I would highly appreciate if the authors can provide the basis for some of the parameters for the leaching experiments: for e.g., for the major ion leaching experiments, authors have taken 0.18 g ash with two size fractions (< 63 ïA∎m and 250-1000 ïA∎m). What is the basis for using these leaching experiment parameters? Also, I noticed some discrepancies in connection to the leaching experiments: earlier in the manuscript in the methods sections, the authors described the leach experiments for trace metals (dFe) and Fe(II) to be very similar, however, later in the manuscript, the authored described both the leaching experiments as not comparable.

Detailed comments following the ms structure:

Line 81: Insert "deposition" between "ash" and "on".

Lines 86 to 90" Rephrase the sentence as "In contrast, there are several adverse effects

of ash deposition on marine organisms that include: (i) metal toxicity (Ermolin et al., 2018), especially under high ash loading, and/or (ii) ingestion of ash particles by filter feeders, phagotrophic organisms or fish (Newcomb and Flagg, 1983; Wolinski et al., 2013)".

Lines 92 to 94: Rephrase as: suggesting that "significant ash deposition on aquatic environments can also impact and perturb their carbonate system."

Line 96: Insert "the" between "to" and "source".

Line 96: Insert "abundance of" between "where" and "macronutrients".

Lines 99 to 100: Rephrase as "In contrast to the 2013 Eyjafjallajökull plume over the North Atlantic, the 2015 ash plume over the region from the Calbuco eruption. . . . . .".

Line 104: Replace "of" by "from".

Line 121: Looking at the mean monthly river water flows, the Puelo river looks to be bigger/major than the Petrohué River.

Line 127: Replace "marine primary production high" with " high marine primary production".

Figure 1: Please label the scale in C.

Figure 1: Can you provide the areal extent of ash deposition from the eruption of Calbuco volcano in the figure.

Figure 1: Please show the location of Cochamó on the map.

Line 148: Can you kindly elaborate on what is meant by the surface of a plastic container?

Line 151: Please provide the location from where the south Atlantic seawater sample was collected.

Line 157: What was the duration for Mucasol stage?

Lines 167 to 168: Replace " shaken by hand" with "manually shaken".

Line 167: Replace "into" with "to".

Lines 174 to 175: How was the instrument calibrated. Could you please provide some more details?

Line 175: Replace "dissolution" with "leaching".

Line 179: Replace "dessert" with "desert".

Line 183: Insert "major" before "ions".

Line 184: This is the only place in the manuscript where any sample from the Aysén Fjord is mentioned.

Lines 191 to 193: Rephrase as: "Samples were immediately analysed for total alkalinity (AT) via a potentiometric titration. . ...".

Line 200: Please expand APHA.

Lines 203 to 206: At what station/location were these measurements made?

Line 204: How was the dissolved oxygen sensor calibrated?

Line 230: Replace "onto" with "of sampled water through".

Line 232: Replace "was" with "were".

Line 247: Could you provide a reference for significant spread of 2015 Calbuco ash to Pacific and Atlantic regions.

Line 269: Please provide the location of mooring station here. It has been mentioned later in line 311.

Figure 4: The May 16 diatom abundance is very high in two extreme stations in fjord: stations A and C, while it is lowest in station B (intermediately placed in the fjord). Can authors explain this?

Lines 303 to 305: As the data plotted on figure 4 shows, the lower diatom abundances were observed in middle station B (open circle) around May 15.

Figure 5: If discussing the brackish water leach experiments at first, place the results for brackish water on the left-hand side panel.

Table 1: It was earlier mentioned by the authors that for leaching experiments for major ions, both brackish water and deionized water were used. Table 1 only presents data for deionized water. Where are the results for leaching experiments with fjord brackish waters?

Figure 6: One of the data points (on all plots) at high ash addition (between 5 and 6 mg) has error in x-data (ash, mg). I assume the ash loading/addition was based on precise weight of ash added to test waters, so it must ne known well.

Figure 6: Here, the authors have mentioned that effect of trace metal leaching upon ash addition was estimated by deducting the initial seawater trace metal seawater concentrations. This has not been mentioned in methods sections. Please provide these details in methods section.

Lines 393 to 395: This is an important point, should have been brought out earlier.

Lines 496 to 497: This is the first time the authors have discussed the relative impact of ash fallout on their stations in Reloncaví fjord. Can the authors discuss this earlier in the manuscript (in section 2)?

Lines 575 to 577: The south western Atlantic chl-a data also shows some significant excursions close to the Calbuco eruption. Also, once the chl-a dips to lowest values close to the Calbuco eruption, it again increases around April 25, concomitant with an increase in aerosol index. Is this due to atmospheric transport of Calbuco ash and its deposition over the region?

Please also note the supplement to this comment:

[Figure]

https://os.copernicus.org/preprints/os-2020-65/os-2020-65-RC1-supplement.pdf

---

## Referee Comment (RC2) · Pierre Delmelle (Referee) · 31 Aug 2020

The manuscript presents the results of a study aimed at assessing the impact of ash deposition on marine primary productivity in Reloncaví fjord, Patagonia in the wake of the 2015 eruption of Cabulco volcano, Chile. The authors performed ash leaching experiments using deionised water and seawater, and measured the releases of Fe(II), total dissolved Fe and other trace metals, including Co, Ni, etc. They also report

in situ measurements of diatom abundances in Reloncvai Fjord, and satellite-derived chlorophyll-a concentration for the ocean region (SE Pacific) affected by ash deposition. The main authors' conclusions are: (i) the increase in diatom abundance inferred to have occurred in the Reloncavi Fjord may relate to ash inputs, but other factors can also explain this observation; (ii) a short duration phytoplankton bloom in the SE Pacific which coincidess in time and space with the ash cloud dispersion; (iii) a decrease with time in the Fe(II) (the most bioavailable iron redox species) release from ash, suggesting that "fresh" ash may act as a stronger source of Fe(II) than previously thought.

I my view the ms of Vergara-Jara et al. requires substantial work. I have found the ms disjointed and it is difficult to see a coherent story. It contains many vague statements that are not backed up by a careful analysis of the data. For example, the authors conclude that Fe(II) release decreases from dry ash samples over time ('aged ash') and emphasise that it is a key result of their study. However, this trend is not apparent from the dataset presented; Figure 7 shows that several measurements corresponding to different ash "ages" produce the same Fe(II) release. A more careful and quantitative analysis is required.

The study also includes measurements of other ash Etna and Chaiten) and dust (Saharan dust, glacial flour, iceberg-borne particles) materials. The reason for selecting these samples is obscure if not random. The data acquired in relation to these samples are almost not used in the discussion.

While recent studies address the question of the solubility of Fe(II) and Fe(III) in volcanic ash (Maters et al., 2016; Maters et al., 2017), these are not used here. The authors also barely acknowledge previous works on metal release from ash (Hoffman et al, 2012 and subsequent studies), although this information could be used for comparison purposes. In general, a more careful analysis and use of the scientific literature on the subject is needed.

I also had some difficulties with the different methods utilised to assess the element release from the ash and other airborne materials. The fresh and brackish water leaching experiments were carried out using two ash size fractions ($< 63 \, \mu$m and 250-1000 $\mu$m). What does justify the choice of the two size fractions, except perhaps the availability of sieves in the laboratory? How well these size fractions represent the actual ash deposits? What is the corresponding specific surface area of the ash particles in each size fraction? The leaching protocols differ between the different measured parameters. This does not allow the rigorous testing of potential relationships between different ash properties.

Overall, the ms suffers in places from gloss simplifications and insufficient use of the huge literature body existing on the processes controlling the solubility of Fe in airborne mineral particles. The discussion is wobbly and the authors jump to conclusions quickly, although the data do not clearly support them.

Section 4.1 "Local drivers of 2015 bloom dynamics in Reloncavi Fjord" is a list of the potential factors that could explain the observed bloom and this section could have been written up without any prior data. I do not think it adds any new knowledge that would contribute to improve our understanding of the potential impact of ash inputs to (marine) water bodies.

Attributing Fe release from ash exclusively to the presence of iron-bearing salts is misleading. Leaching and dissolution of the aluminosilicate glass and minerals contained in ash is also a source of Fe(II) and Fe(III) (see Maters et al., 2017; 2017 and other studies).

L39 – should be cloud not plume. Same applies throughout the ms

L48 F not Fl but F. Same applies throughout the ms and tables

L50 "higher than usual" by how much? Two times? An order of magnitude? Outside the multiannual variability?

L52 You should be transparent in the abstract that this is highly speculative (since no

other measures in the fjord point to a phytoplankton response to ash addition).

L62 How can a micromolar concentration of Fe(II) be released when above you say only nanomolar concentrations of Fe are released?

L64 This is not justified. First, you assume only Fe(II) is bioaccessible, and second, it is based on Fe(II) decreasing in aged ash which is not well supported by the data.

L67 meaning?

L77 2010 not 2013

L81 Why 'therefore'? The preceding sentence deals with a case that is not in a HNLC area of the ocean.

L94 Perhaps in fresh water but I imagine any shift in seawater pH is extremely transient and localised due to strong buffering!

L99 2010 not 2013

L148 Had the ash been rained on in the interval between the eruption and sampling?

L151 But South Atlantic seawater is presumably not HNLC water, so its properties might lead to findings here only relevant to ash input to the South Atlantic. For example, HNLC seawater might have different types/abundances of Fe-binding ligands.

L152 Is there a basis for this range? Does it mimic the Calbuco ash loading to the SW Atlantic and SE Pacific regions studied?

L167 Again, if South Atlantic seawater is not Fe-limited, then presumably measurements of Fe(II) concentrations in this water cannot be generalised to reflect Fe release behaviour from ash in Fe-limited seawater. For instance, different types/abundances of Fe-binding ligands in HLNC water might strongly influence dissolved Fe concentrations on ash input to seawater.

L168 Why is the ash loading concentration different than that used for the trace metals

above?

L175 What 'dissolution experiments'? Do you mean ash leaching in seawater?

L179 Should be desert and please specify which, for consistency with specifying the ash source (Etna, Chaiten).

L180 Please specify sources

L183 Please revise heading - species responsible for alkalinity, ions, and nutrients are not mutually exclusive.

L184 I thought it was Reloncavi Fjord?

L187 Do you mean deionized water here? If so, please say this instead to avoid misinterpretation to mean environmental/fjord fresh water.

L188 0.18 g ash in 40 mL is 4.5 g/L or a 1:222 ash:water ratio. Where does this come from (it is not from Witham et al. 2005)?

L200 Were saturation indices calculated for all species in solution?

L222 This should come after 'at 3 depths'.

L328 Isn't it 4.5 g/L (0.18 g in 40 ml)?

L335 This is probably just a surface area effect (i.e. smaller size particles for the same mass of ash release correspondingly more ions due to the greater surface area in contact with solution)

L335 Leaching not dissolution.

L338 Because CaSO4 salts are not as soluble?

L350-352 I don't see support for this statement. From Figure 7 (right) showing all replicates, the couple of high Fe release values per unit ash mass seem like outliers. In fact two of the lowest ash loadings exhibit among the lowest Fe release per unit

mass.

L356 Be careful with these statements, there is a lot of overlap of error bars so any apparent increase may not be significant.

L370 Why this temperature and not room temperature, like the others?

L374 If not in the legend, at least here in the caption you should state what they are (desert dust, glacial flour, iceberg particles) etc.

L374-375 It seems that the two other volcanic ash (Etna, Chaiten) are included here to represent older ash samples, but if that's indeed the purpose, this is not a valid comparison because the different Fe chemistries (total Fe content, Fe redox speciation, and Fe mineralogy) in these samples are likely to be greater drivers of their Fe(II) release behaviours than the different ash ages. This must be acknowledged, or else I suggest removing the Etna and Chaiten ash from this study altogether.

L381 The Figure 7 y-axis reports Fe(II) release in nmol/g. Please be consistent for clarity.

L391 Perhaps only at low ash loadings, it's hard to say from the few data points for 9 month old ash. I would not consider this a clear trend at all, and in fact the Fe(II) release from 4 month old ash is often higher than from 2 week old ash at the same ash loading.

L393 This is not at all supported by the data shown in Figure 7 (left), see my comment above.

L397 Are you sure that this corresponds to volcanic ash and not to volcanic sulphate aerosol?

L511 Again, have you tested for saturation of Fe(II) (and other species) in your leachates? It would be useful to explore the possibility of secondary phase precipitation explaining decreasing dissolved Fe(II) with increasing ash loading.

L512 Again, I don't see clear evidence for this statement in the data (e.g., Figure 7 right).

L520 Although this notion is propagated in the literature, there remains a paucity of evidence for Fe salts on ash surfaces! This section is missing important information on the forms of Fe (Fe(II) and Fe(III) in ash) - in aluminosilicate glass and mineral network and Fe(-Ti) oxide minerals. Will mislead readers to claim Fe salts are responsible again when we know that's not the case.

L522-524 No. Fast release of Fe(II) is more likely to originate from leaching of the aluminosilicate glass. Please do your homework (e.g., see Maters et al. 2016 and 2017 - those studies done at pH 1, 2 and 5 but are still relevant sources of info on the forms of Fe in ash and its release into solution)!

L525 See earlier comments, this statement is simply not well supported by the available data.

L527 There's no such thing as acidic surface coatings. The presence of any salts on ash surfaces is the end product of prior reaction between acids (H2SO4, HCl) and the aluminosilicate –> neutralisation

L529 How would an acid-base reaction be responsible for Fe(II) conversion to Fe(III)? Presumably the Fe(II) at the ash surface has somehow been oxidised to Fe(III) during storage, or else made to be less mobile in some other way...

L530 Again, based on the fact that your data do not support the conclusion that aged ash releases less Fe(II), this statement should be removed. In any case, the term 'aged' in the mineral dust/glacial flour/volcanic ash community often refers to material that has interacted with other species during atmospheric transport ('aging'), if anything increasing Fe solubility and Fe(II) mobilisation over time (e.g., see Maters et al. 2016).

L532 This is true simply because we know that airborne material (dust, flour, ash) undergoes atmospheric processing, including exposure to inorganic and organic acids

and cloud condensation and evaporation cycles, that is likely to modify the Fe solubility and speciation in the material before deposition to water bodies. Please acknowledge the huge body of literature in this area.

L536-544 All this should be removed because it surrounds a claim about aging/Fe(II) release that is not supported by the data here.

L552 What aerosols? Please specify since the Fe chemistry in different particulate materials can vary drastically.

L553 Specify Calbuco 2015 ash. Saying 'fresh volcanic ash' is a gross generalisation and completely neglects existing studies reporting variable Fe release from ash, including Fe(II) and Fe(III) release by ash from different eruptions - Maters et al. 2017. Okay that study done at low pH but it shows that Fe chemistry in ash is highly variable and likely plays an important role in Fe release from ash in solution.

L561 Please specify ash or particle, if that's indeed what the satellite detected. The term 'aerosol' in the volcanology context most often refers to sulphate aerosol.

---

## Referee Comment (RC3) · Anonymous Referee #3 · 10 Sep 2020

General Overview

The manuscript by Vergara-Jara et al addresses some important topics with respect to the influence of volcanic eruptions on marine productivity – notably the release of major ions and trace elements from ash deposited to surface waters and the range of possible biological responses to those inputs, based on biogeochemical conditions present at the time. The authors discuss results from a series of leaching experiments performed

on ash from a 2015 eruption of the Calbuco volcano and combine this information with time-series data from the nearby Reloncavi Fjord to try and identify possible influence from ash inputs. They also use satellite data of ash dispersion from the eruption and marine chlorophyll-a concentrations to compare responses of ash inputs from the volcano to the eastern South Pacific and the western South Atlantic Oceans. Overall, I thought that the manuscript was well written and fairly concise, though there are places where it would benefit from a more in-depth discussion of the caveats associated with the data. The manuscript addresses phenomena that it is difficult to plan comprehensive studies of and could therefore be a useful addition to the literature on these topics. However, there are issues with the current version that need to be addressed before publication, which I will lay out below.

Specific Comments

The ash sample was collected two weeks after the eruption and the meteorological data in Figure 3 indicates some rainfall (albeit in a different location to where the ash was collected) in the period between eruption and sample collection. Is it possible that the collected ash had been exposed to rainfall before collection? If so, can the authors discuss how this may have influenced their findings in terms of leachable trace elements and major ions?

There are numerous points to address with the methodology of the leaching experiments:

1) The leaching experiments conducted for determination of dissolved TMs into seawater, Fe(II) into seawater, and alkalinity and major ions into brackish and deionized water all use different experimental approaches, in terms of volumes used, ash loading, and length of mixing time. This is most relevant for the comparison of dissolved Fe versus Fe(II). While such differences in approach are sometimes unavoidable, the authors should at least discuss the potential for complications resulting from these differing approaches, particularly for the iron data – can they rule out any methodological

artefacts in the data?

2) The method for the Fe(II) release leaching experiment states that subsamples were introduced into the flow-injection system without filtration. Does this not admit a potential positive bias in the released Fe(II) data through small ash particles getting trapped in the FIA manifold and undergoing further leaching and/or reaction with the FIA reagents?

3) In the Fe(II) method description it states that measurements were made every two minutes for 30 minutes for each ash loading, and that the data presented are "mean concentrations measured from 2-30 minutes after adding ash into solution". Does this mean that all of the data from 2-minute intervals are averaged to produce the data points in Figure 7? Was there no significant temporal progression of concentration over this ∼30-minute period? A related point is that if the data points in Figure 7 are mean values, presumably the standard deviations could be added to give a clearer idea of the significance of differences between datapoints.

4) Is there any scientific significance to the two ash size fractions chosen in the alkalinity/major ions leach experiments?

5) The description of dFe release in section 3.4 is described as being most efficient at the lowest ash loading per unit volume of seawater (line 351). Similarly, in Section 4.2, release of Fe and Mn is referred to in terms of nmol/g. This would be easier for the reader to visualize if Figure 6 was altered. Either additional plots could be included to plot each element as nmol/g released versus ash concentration in mg/L (as in Figure 7), or these plots could be superimposed on the existing plots by including secondary x- and y-axes.

6) For the Fe data in Figure 6, the value for the lowest ash addition has a large standard deviation. In Figure 7 we see that this is due to one replicate with a very high amount of Fe released per gram and two replicates with low values. The difference is very striking. Can the authors comment on the likelihood that the high value is an outlier

and/or due to sample contamination? If not due to contamination, could this value be due to a methodological artefact?

7) At line 381 the release of Fe(II) from ash is referred to in terms of nmol/L, but Figure 7 does not show this as it relates to ash loading. It may be useful to include an additional panel in Figure 7 that shows the nM Fe(II) release as a function of ash added.

8) At line 393, the authors mention an apparent decrease in Fe(II) release with aging, but the only notable decrease seems to be between 4 months and 9 months, with little apparent difference between the trends at 2 weeks and 4 months. This should be clarified (e.g. "The release of Fe(II) from ash therefore appeared to decrease with aging after several months").

Figure 4 – In section 3.2, this figure is used to make a comparison between diatom abundance at stations in the upper part of the fjord, and historical data from Reloncavi Sound, which presumably undergoes more circulation and has a shorter residence time for waters. In addition, the new data is integrated over the upper 15m, compared to the upper 10m for the literature data. The comparison is quite striking. I appreciate that there is a desire to put the new data into some kind of historical context, but I think the authors should include the caveat in section 3.2 that the new and historical datasets may not be directly comparable. The authors do state in the discussion that the data is not directly comparable to the historical data (lines 426-427), but I think this point also needs to be made in the results section.

Figure 8 – The apparent differing responses in the eastern Pacific and western Atlantic to ash deposition is a very interesting aspect of this study. However, I believe it would strengthen the findings of this paper if the authors could rule out the possibility that the observed response in chlorophyll in the Pacific Ocean is coincidental to the ash input. Figure 8G compares the satellite-derived aerosol index and chlorophyll-a concentration in the Pacific region over which the ash cloud passes. Have the authors looked at making a chlorophyll-a time-series at a similar area that did not see a strong variation

in the aerosol index (for example, the area immediately to the south of the box used in 8G)? If no chlorophyll-a bloom corresponding to that in 8G is observed at this "no-ash" site, it would strengthen the argument that ash deposition was the trigger. Similarly, on the Atlantic side, there is a smaller ash-impacted area to the south of Rio de la Plata evident in panels C and D. Have the authors looked for any possible chlorophyll-a signal in that region and if they have, do the findings concur with the findings in panel H (i.e. that there is no ash-driven bloom)? Admittedly the aerosol index for this area looks substantially smaller, and cloud cover in the later time-period covered (panel F) may prevent a proper analysis of this area. Line 557 - The Browning et al (2015) reference suggests that in some cases, ash can bias satellite-derived chlorophyll-a measurements upwards significantly due to the optical properties of the ash and the algorithms used to convert data into chlorophyll concentrations. Can such a bias be ruled out in this study?

Technical comments:

Line 71 – no need for hyphen in micronutrient

Line 179 and throughout Supplementary Table 1 – replace "dessert" with "desert".

Line 187 – "fresh water" – use deionized water throughout. There is potential for this to be confused with river water.

Line 194 – change to "a reproducibility of <2 umol/kg"

Lines 303-305/Figure 4 – It appears that the legend for Figure 4 is incorrect. It looks as though diatom abundance is greater at stations A and C, rather than B and C as stated in the text. The data in supplementary table 2 indicates that the figure is wrong, rather than the text. Based on the supplementary table, I would say that circles are station A, triangles are station B, and squares are station C.

Line 304 (and 431) – in both cases it is stated that diatom abundances were measured on 16th May, yet the supplementary table gives the date as 14th May. Which is it?

Lines 305-307 – It would be more accurate to say that highest measured chlorophyll was on 30th April at a station close to station C. Based on Figure 4 it can't be said that concentrations decreased to much lower concentrations in June, as there isn't any data shown for June.

Line 339 – No need for "and" after NH4.

Table 1 – It states in the caption that all values are means. It would be more informative to also include standard deviations in the table if the data is from replicates – this would allow readers to assess whether changes observed with time are significant or due to noise in the measurements. Also, how is the detection limit arrived at? Is it 3x standard deviation of a blank?

Line 356 – I'm not convinced that Ni shows that trend – only two additions seem to give a positive increase in Ni concentration, with one of those being the highest ash loading, and this gives a false impression that there is a positive trend. I would group Ni with Cd rather than Co and Pb.

Figure 7 – The "ash 9 months" data does not match that in the supplementary table, in that in the table all four data points are between 18-31.9 mg/L. with corresponding nmol/g values of ∼2 to ∼16. Line 439 – change to "ash deposition per unit area"

Line 509/10 – It seems more appropriate here to refer to Figure 7 (right hand panel), as that shows the data in terms of nmol/g, as mentioned in the text, rather than the nmol/L change shown in Figure 6.

Line 549-550 – is there a possibility here that small particulates could have contributed to the Fe(II) concentrations (as these samples were not filtered between ash addition and analysis)? See specific comment earlier in review.

Line 605 – rather than "correlation", which suggests a statistical relationship between the two parameters, I would suggest rephrasing this to something more general, such as "atmospheric ash loading was related to an increase in chlorophyll-a" (that is unless

the authors can include a panel in figure 8 that does indeed show a correlation between satellite derived chl-a and aerosol index). Note also that the ash distribution shown by the aerosol index does not necessarily translate to "deposition" as stated here.

---

## Author Comment (AC1) · 20 Nov 2020

Q. Questions/comments R. Responses

With respect to the major concerns: changes have been made to the section 2 (Materials and methods), including a table with information from all the leaches conducted.

Q. As is in the present version of the manuscript, section 2 (Materials and methods)

[Figure]

is difficult to follow. When I started reading this section, I was looking for a common subsection detailing all the samples (and their sampling location) for the present study. Opening section 2, subsection 2.1 is placed well and describes the study area providing the regional details. However, I would like to suggest the authors insert a subsection 2.2, providing details of all the samples collected and analysed in the present work. In addition, there are some minor concerns with this section (section 2). For e.g. (i) this study is based on the changes in biogeochemistry of Reloncaví fjord and the Atlantic Ocean immediately after the eruption of Calbuco volcano. However, some samples from another location (Aysén fjord) were also discussed in the manuscript (line No. 184). This location has only been mentioned here and nowhere else in the manuscript. (ii) As mentioned in line 134, there is a mooring station located in Reloncaví fjord, its location is not known until late in the manuscript (Line 311). R. For Aysen fjord, we have highlighted in section 2.3, we changed the first paragraph to clarify why we were using that water and not Reloncaví fjord water (new lines 205-207). In simple terms, during our first visits to the main fieldsite (Reloncaví) the ash loading in water was extremely high, and we were concerned that even in the weeks after the main ash load diminished there may have been a legacy of ash in the water composition. It therefore made little sense to use Reloncaví water for ash leaches, so we collected water from a nearby system where any ash deposition was negligible.

Q. I have some concerns regarding the leaching experiments done by the authors. As is the case in the present work, different leaching experiments were performed for different set of parameters (major ions, trace metals and Fe(II)). Why did the authors use different leaching procedures for different parameters in the present work? It was the same ash falling over the waters of the fjord and Pacific Ocean. So, to see the combined effect of the ash falling on fjord waters, authors should have used similar leaching protocols for major ions, trace metals and Fe(II). Also, the authors have used deionized water for their leaching experiments for major ions. Why did the authors not consider using the trace metal free seawater for their leaching experiments? Also, some of the methods are not clear. For the leaching experiments for major ions, the authors

have mentioned correcting the abundances for initial water concentrations. Was this also done for trace metal leaching experiments? The authors have mentioned that for leaching experiments for major ions, they used both fjord waters and deionized water, but table 1 only provides results for leaching with deionized water. Also, I would highly appreciate if the authors can provide the basis for some of the parameters for the leaching experiments: for e.g., for the major ion leaching experiments, authors have taken 0.18 g ash with two size fractions (< 63 ïA■m and 250-1000 ïA■m). What is the basis for using these leaching experiment parameters? Also, I noticed some discrepancies in connection to the leaching experiments: earlier in the manuscript in the methods section, the authors described the leaching experiments for trace metals (dFe) and Fe(II) to be very similar, however, later in the manuscript, the authored described both the leaching experiments as not comparable. R. Different procedures are necessary for two reasons. First, with respect to Fe(II), Fe(II) has a short half-life at room temperature (> 1 minute in seawater). It takes at least one minute to measure the concentration using the most rapid available method (flow injection analysis with a continuous flow of sample). Thus any standard leaching protocol at room temperature will invariably measure very low levels of Fe(II) irrespective of whether any was released from ash, or not. Consequently, for Fe(II) analysis we ran experiments at low temperature. For remaining trace metals we opted for a room temperature leach to follow prior work. The leaches are therefore roughly comparable as both test the effect of ash addition on dFe release in seawater, but in the standard protocol the initial 'pulse' of Fe(II) detected in the chilled experiments has likely already decayed to some extent which cannot be quantified.

The condition of coastal seawater and offshore seawater are always (broadly speaking) very different with respect to trace element concentrations; much higher concentrations are present in coastal waters. The amount of metal that can be leached from ash into solution is very sensitive to the ambient concentration of Fe already present. For comparability with prior work, and to focus on the potential effect of ash on offshore trace metal dynamics (which is where any metal-fertilization would be expected to be most

evident) we therefore used offshore seawater. The difference between TA and DIC anywhere in the ocean is far less pronounced than the difference in metal levels. Note that the DI experiment provides additional opportunity for comparison with similar studies (e.g. Jones and Gislason 2008), in addition to being more sensitive to the estimation of net leaching of major ions and macronutrients. Similarly, for trace elements, saline leaches are the most appropriate for our research question as Fe is not generally considered an important control on freshwater productivity, nor would it likely increase the net loading to coastal ecosystems from riverine inputs.

For the analyses of the data from experiments explained in section 2.3, the correction of the abundances for the initial water concentrations was done in order to focus on the ion inputs from the ashes and not in the total ion content from all the leachates. Because we were working with fresh water, we have to subtract the initial ion concentration after the experiment was done. Similarly for trace metal leaches, the initial concentrations in seawater are noted and deducted where concentrations are plotted as the change in concentration before/after ash.

The different ash size fraction used in section 2.3 were made in order to quantify accordingly the impact from different particles from the same sample, following the recommendations of Witham et al., 2005. The main constraint was the amount of ash available. Note that although the total mass of unhydrated ash was limited, the ask/water ratio for DI leaching experiments was of the same order of magnitude as for Jones and Gislason 2008. Note also that the single 45-125 um size fraction used in J&G's experiment, although standardized across sites, was most likely selected based on the need to maintain circulation through a continuous flow reactor. The two size fractions used here represent standard mesh sizes, and were chosen for the principal purpose of demonstrating the proportionately greater effect of the fine fraction selecting the finest and coarsest fraction which we would still have enough material to work with. Finally, we note that grain size distribution will vary widely across any given ash plume, being the overriding factor in terms of comparing effects across eruptions.

Q. Line 81: Insert "deposition" between "ash" and "on". R. Line 81: recommendation accepted.

Q. Lines 86 to 90: Rephrase the sentence as "In contrast, there are several adverse effects of ash deposition on marine organisms that include: (i) metal toxicity (Ermolin et al., 2018), especially under high ash loading, and/or (ii) ingestion of ash particles by filter feeders, phagotrophic organisms or fish (Newcomb and Flagg, 1983; Wolinski et al., 2013)". R. Lines 86 to 90: Now the new paragraph is as follow: "In contrast, apart from inducing light limitation, there are several adverse effects of ash deposition on marine organisms that go from metal toxicity 1- particularly under high dust loading 2- or more generally from the ingestion of ash particles by filter feeders, phagotrophic organisms or fish 3,4".

Q. Lines 92 to 94: Rephrase as: suggesting that "significant ash deposition on aquatic environments can also impact and perturb their carbonate system." R. Lines 92 to 94: suggestion accepted.

Q. Line 96: Insert "the" between "to" and "source". R. Line 96: suggestion accepted.

Q. Line 96: Insert "abundance of" between "where" and "macronutrients". R. Line 96: suggestion accepted.

Q. Lines 99 to 100: Rephrase as "In contrast to the 2013 Eyjafjallajökull plume over the North Atlantic, the 2015 ash plume over the region from the Calbuco eruption......". R. Lines 99 to 100: suggestion accepted.

Q. Line 104: Replace "of" by "from". R. Line 104: suggestion accepted.

Q. Line 121: Looking at the mean monthly river water flows, the Puelo river looks to be bigger/major than the Petrohué River. R. Line 121: It is. Now the rivers are mentioned in order of their flow.

Q. Line 127: Replace "marine primary production high" with "high marine primary production". R. Line 127: suggestion accepted.

Q. Figure 1: Please label the scale in C. Can you provide the areal extent of ash deposition from the eruption of Calbuco volcano in the figure. Finally, please show the location of Cochamó on the map. R. Figure 1: Figure 1 now has the scale labeled and the location of Cochamó town is marked on the map. Also, a new reference extent of the ash cloud is showed in Fig. 1 C (although note that the ash cloud changed from day-to-day). Now this new information is clarify in the text in section 2.1 at the end, the following paragraph was added: "The Chilean Geological-minning Survey (Servicio Nacional de Geología y Minería, SERNAGEOMIN) elaborated daily technical reports with information about the area of dispersion for the emitted ash (http://sitiohistorico.sernageomin.cl/volcan.php?pagina=4&iId=3). We used this information to create a reference aerial extent of ash deposition for the days after the eruption (Fig. 1, C)."

Q. Line 148: Can you kindly elaborate on what is meant by the surface of a plastic container? R. Line 148: A plastic tray which we lined with LDPE plastic that is typically used to wrap samples for trace metal analysis due to its low trace metal content.

Q. Line 151: Please provide the location from where the south Atlantic seawater sample was collected. R. Line 151: suggestion accepted, South Atlantic (40°S – it was filled pumping underway to keep the inflow clean and there is no fixed longitude, but the exercise was conducted offshore)

Q. Line 157: What was the duration for Mucasol stage? R. Line 157: Clarification made (3 days), now more details have been added to the text

Q. Lines 167 to 168: Replace "shaken by hand" with "manually shaken". R. Lines 167 to 168: suggestion accepted.

Q. Line 167: Replace "into" with "to". R. Line 167: suggestion accepted.

Q. Lines 174 to 175: How was the instrument calibrated. Could you please provide some more details? R. Lines 174 to 175: Standard additions of Fe(II) were used to

calibrate the instrument, with peak height then used to derive Fe(II) concentration. Additional details are added (new lines 188-196).

Q. Line 175: Replace "dissolution" with "leaching". R. Line 175: suggestion accepted.

Q. Line 179: Replace "dessert" with "desert". R. Line 179: suggestion accepted.

Q. Line 183: Insert "major" before "ions". R. Line 183: suggestion accepted.

Q. Line 184: This is the only place in the manuscript where any sample from the Aysén Fjord is mentioned. R. Line 184: A clarification is made why we used this water (to avoid using water which had already had a high ash load in our leach experiments)

Q. Lines 191 to 193: Rephrase as: "Samples were immediately analysed for total alkalinity (AT) via a potentiometric titration….…". R. Lines 191 to 193: suggestion accepted.

Q. Line 200: Please expand APHA. R. Line 200: APHA explanation extended. The full name: American Public Health Association, 2006.

Q. Lines 203 to 206: At what station/location were these measurements made? R. Lines 203 to 206: Clarification made. Time series is from the oceanographic buoy at the Reloncaví fjord.

Q. Line 204: How was the dissolved oxygen sensor calibrated? R. The sensor installed at the NPOB was factory calibrated and was a brand-new instrument when installed. Details are in Vergara-Jara et al., 2019.

Q. Line 230: Replace "onto" with "of sampled water through". R. Line 230: suggestion accepted.

Q. Line 232: Replace "was" with "were". R. Line 232: suggestion accepted.

Q. Line 247: Could you provide a reference for significant spread of 2015 Calbuco ash to Pacific and Atlantic regions. R. Line 247: Two important references have been added.

Q. Line 269: Please provide the location of mooring station here. It has been mentioned later in line 311. R. Line 269: The Reloncaví fjord mooring is clearly label in Fig. 1 C. As North Patagonia Oceanographic Buoy.

Q. Figure 4: The May 16 diatom abundance is very high in two extreme stations in fjord: stations A and C, while it is lowest in station B (intermediately placed in the fjord). Can authors explain this? R. Figure 4: There's no clear reason on why station B at the center of the sampled locations got those values, but, it is not unusual to see patchiness in biological parameters (chla, or group distributions) along the fjord presumably due to local circulation patterns that can create zones with different resident time inside the fjord.

Q. Lines 303 to 305: As the data plotted on figure 4 shows, the lower diatom abundances were observed in middle station B (open circle) around May 15. R. Lines 303 to 305: Correction made following suggestion. Peak was at stations A & C around May 16, and lower abundances was at station B (open circle) as well noticed by this reviewer.

Q. Figure 5: If discussing the brackish water leach experiments at first, place the results for brackish water on the left-hand side panel. R. Figure 5: The figure now shows the brackish water at left, and the scale has been fixed in order to show better the differences between the treatments.

Q. Table 1: It was earlier mentioned by the authors that for leaching experiments for major ions, both brackish water and deionized water were used. Table 1 only presents data for deionized water. Where are the results for leaching experiments with fjord brackish waters? R. Table 1: The fresh fjord brackish water used in the experiments has a relatively high background concentrations of macronutrients and major ions, so for major ions we present only results from DI water where it was possible to observe the increase from the ash leachate.

Q. Figure 6: One of the data points (on all plots) at high ash addition (between 5 and

6 mg) has error in x-data (ash, mg). I assume the ash loading/addition was based on precise weight of ash added to test waters, so it must be known well. R. Figure 6: The vertical error bar shown is the estimation of the standard deviation as the figure legend explains. Horizontal bars reflect the small variation in ash mass between replicates. Ash was pre-weighed, and then following addition to the experimental solution the vials were re-weighed to determine the exact mass added to solution – static charges have resulted in a low, but varying, loss of sample during transfer, representing one of the challenges in working with small quantities under trace metal clean conditions.

Q. Figure 6: Here, the authors have mentioned that effect of trace metal leaching upon ash addition was estimated by deducting the initial seawater trace metal seawater concentrations. This has not been mentioned in methods sections. Please provide these details in methods section. R. Figure 6: The initial concentration was stated and we also stated that concentrations were presented as the change compared to initial concentrations in seawater. For clarity we add this information again (new lines 200-203).

Q. Lines 393 to 395: This is an important point, should have been brought out earlier. R. Lines 393 to 395: Note following more careful consideration that there is no clear decline in the Fe(II) released between 2 weeks and 4 months after collection, we have removed this section.

Q. Lines 496 to 497: This is the first time the authors have discussed the relative impact of ash fallout on their stations in Reloncaví fjord. Can the authors discuss this earlier in the manuscript (in section 2)? R. Lines 496 to 497: Suggestions accepted, a new brief sentence has been added to the text in section 2.

Q. Lines 575 to 577: The south western Atlantic chl-a data also shows some significant excursions close to the Calbuco eruption. Also, once the chl-a dips to lowest values close to the Calbuco eruption, it again increases around April 25, concomitant with an increase in aerosol index. Is this due to atmospheric transport of Calbuco ash and its

deposition over the region? R. Lines 575 to 577: This is certainly a possibility, although the peak in chlorophyll-a prior to the increase in UV aerosol index makes it difficult to ascribe the subsequent peak to ash fertilization. We have however now revised the manuscript to state:

Conversely, ash deposition into the south western Atlantic indicated by the UV aerosol index did not lead to such a clear corresponding change in chlorophyll-a concentrations (Fig. 8H), although with the available data it is not possible to rule out the possibility of fertilisation completely (e.g., whilst also being proceeded by a larger chlorophyll-a peak on August 21st, there is a peak in chlorophyll-a at August 25th coincides with elevated UV aerosol index).

———————————————————

---

## Author Comment (AC2) · 20 Nov 2020

Q. Questions/comments R. Responses

Q. It contains many vague statements that are not backed up by a careful analysis of the data. For example, the authors conclude that Fe(II) release decreases from dry ash samples over time ('aged ash') and emphasise that it is a key result of their study. However, this trend is not apparent from the dataset presented; Figure 7 shows that

several measurements corresponding to different ash "ages" produce the same Fe(II) release. A more careful and quantitative analysis is required.

R. For the Fe(II) concentrations, because of the parametric fit to these data to ash loading and the fact they weren't collected specifically to test aging (we noticed afterwards there may have been some decline) with replicates of the exact same loading, it is difficult to show if there is a trend, especially since we agree that there is no clear change between the first two time points. This being the case we have removed comments concerning the potential decline from the manuscript.

Q. The study also includes measurements of other ash Etna and Chaiten) and dust (Saharan dust, glacial flour, iceberg-borne particles) materials. The reason for selecting these samples is obscure if not random. The data acquired in relation to these samples are almost not used in the discussion.

R. We simply wanted to test if other particles also released Fe(II) upon addition to seawater as there is sparse literature specifically testing this and so tested a broad range of particles from well characterized materials we have in our collection. However, as these are all aged, we agree the comparison is not particularly useful so have removed these parts of the manuscript.

Q. The fresh and brackish water leaching experiments were carried out using two ash size fractions (< 63 _m and 250-1000 _m). What does justify the choice of the two size fractions, except perhaps the availability of sieves in the laboratory? How well these size fractions represent the actual ash deposits? What is the corresponding specific surface area of the ash particles in each size fraction? The leaching protocols differ between the different measured parameters. This does not allow the rigorous testing of potential relationships between different ash properties.

R. We have included a new table that summarize all the leaching experiments done in this research, the analysis, the water used for the leachates, etc. In order to better explain the different methods used. The leaching protocols had to differ to make
some measurements e.g. measurements of major ions in saline waters would not be possible due to the high background level of most ions, measurement of Fe(II) is not meaningfully possible at room temperature etc.

The size fractions used were selected to test the effect from different particle size. The corresponding size fractions used and their respective percentages from the total of the sample are: > 2360 um (4.54%); < 2360 um & >1000 um (6.85%); <1000um & >250um (31.12%); <250um & >125um (24.14%); <125um & >63um (18.04); <63um (15.31%). This information in now also included (new lines 215-217).

The main constraint on the selection of ash size fractions was the total mass of unhydrated ash available for experimentation, which generally is very limited for any given study and in our case precludes the robust testing of all possible experimental treatments. We therefore focused on the most meaningful treatments: major ions and macronutrients are effectively undetectable in given background levels in brackish or seawater matrices, and Fe is probably not a significant factor in terms of biogeochemical effects in freshwater systems. Particle size distributions generally vary considerably over the extent of ash plumes (and to some extent over short time scales), hence there is no standard "actual ash deposits" but a range, which for this event has not been to our knowledge characterized. The size fractions reported here are standard sieve sizes, their relation to previous studies is discussed above and they were selected as fractions which could be sieved and still produce enough material to work with (larger, or smaller, sieves would have not yielded enough material to conduct the work described herein from our bulk ash sample). Although the surface area was not estimated, based on previous studies (Brantley et al 1999 and Gauttier et al 2001, op cit. Jones and Gislason 2008), specific surface area will approximately double for each increment in smaller grain size (again based on standard 500, 250, 125, 63 $\mu$m series). Ultimately, based on the size fractions represented above, we tested both the dominant size fraction by mass (250-1000 um) and by surface area (<63 um), the latter proving an overwhelming contribution to leaching products as demonstrated here.

[Figure]

Q. Overall, the ms suffers in places from gloss simplifications and insufficient use of the huge literature body existing on the processes controlling the solubility of Fe in air-borne mineral particles. The discussion is wobbly and the authors jump to conclusions quickly, although the data do not clearly support them.

R. We have carried out extensive modifications to remediate the gloss simplifications and insufficient use of the huge literature body, following the suggestions from all re-viewers particularly removing the material which speculated a decline in Fe(II) release with aging. We focus on the novel aspects, which as commented by other reviewers, concern the Fe(II) release which has been poorly investigated into seawater.

Q. Section 4.1 "Local drivers of 2015 bloom dynamics in Reloncaví Fjord" is a list of the potential factors that could explain the observed bloom and this section could have been written up without any prior data. I do not think it adds any new knowledge that would contribute to improve our understanding of the potential impact of ash inputs to (marine) water bodies.

R. Section 4.1 is a section that was written thanks to the extensive and exhaustive en-vironmental data gathered in this environment. To our knowledge similar biochemical in situ monitoring datasets are not present in the literature that have fortuitously mea-sured, with high time resolutions (h), the potential effect of a volcanic eruption within an environment of similar characteristics like the Reloncaví fjord. We believe that having the possibility to measure the in situ data of a natural water body should be of interest for different scientific disciplines. We acknowledge that there does not appear to have been dramatic changes in the fjord and thus our summary of the shifts observed- which appear to be largely seasonal and not specifically in response to the ash deposition- is not particularly exciting, but this can only be concluded having seen the data.

Q. Attributing Fe release from ash exclusively to the presence of iron-bearing salts is misleading. Leaching and dissolution of the aluminosilicate glass and minerals con-tained in ash is also a source of Fe(II) and Fe(III) (see Maters et al., 2017; 2017 and

other studies).

R. We have studied the literature concerning the evidence for iron-bearing salts and agree the evidence for this, whilst multi-faceted, is not entirely conclusive. However, our main line of argument, which was not developed previously, is the temporal development of Fe(II) during the leach experiments. This does show a pulse of Fe(II) released in the first minute of dissolution followed by a closer to steady-state situation later.

The studies quoted demonstrate leaching of Fe(II) at low pH. This is not comparable to work conducted in seawater, because the fraction of dFe leached as Fe(II) is sensitive to the solution pH and shows a non-linear relationship with pH (the ferrozine method used to quantify Fe(II) in the above studies is also prone to artefacts under low pH conditions). Similarly, the stability of Fe(II) in solution is highly dependent on pH. Experiments demonstrating that Fe(II) is leached under acidic conditions cannot therefore be used to extrapolate to seawater either from a concentration, or from a mechanistic perspective.

There are numerous papers providing varying lines of evidence for the release of Fe-bearing salts dominating dFe release over short (minutes) time periods. We accept however that these are not conclusive that this is the only source of dFe (or specifically Fe(II)). Following comments from another reviewer, we also now display the Fe(II) data from ash release as a time series (after ash addition to seawater). The temporal trend suggests that much of the Fe(II) is released in the first minute following addition to seawater suggesting that the origin of this Fe(II) is a highly soluble phase. We have amended the section accordingly (new lines 562-574).

Q. Line 39: should be cloud not plume. Same applies throughout the ms R. Line 39: We refer to plume, because is from the eruption itself, while an ash cloud could form afterward from resuspension of old ash deposits. The ash studied here came from the eruption plume. Ash cloud is used in the context of the satellite analyses.

Q. Line 48: F not Fl but F. Same applies throughout the ms and tables R. Line 48: Correction accepted. Changed all throughout the ms.

Q. Line: 50 "higher than usual" by how much? Two times? An order of magnitude? Outside the multiannual variability? R. Line 50: About two times and up to four times higher (Gonzalez et al 2010; Montero et al., 2011) for corresponding season, Autumn).

Q. Line 52: You should be transparent in the abstract that this is highly speculative (since no other measures in the fjord point to a phytoplankton response to ash addition). R. Line 52: We change the phrase to: Within Reloncaví Fjord, average integrated peak diatom abundances were higher than usual by up to two times (May diatom abundance cell*109 m-2), integrated to 15 m depth), with the bloom intensity perhaps moderated due to high ash loadings in the weeks following eruption.

Q. Line 62: How can a micromolar concentration of Fe(II) be released when above you say only nanomolar concentrations of Fe are released? R. Line 62: Note the units are different "nmol l-1" refers to the Fe concentration in solution. "$\mu$mol g-1" refers to the Fe in solution per unit of ash. The units/values are correct as stated.

Q. Line 64: This is not justified. First, you assume only Fe(II) is bioaccessible, and second, it is based on Fe(II) decreasing in aged ash which is not well supported by the data. R. Line 64: This is based on the observation that Fe(II) was released, not on how it aged. A room temperature leach with subsequent measurement of dFe would not detect this Fe(II), it would already have re-oxidised and precipitated as Fe(III) before it could even be filtered and preserved. We are not assuming Fe(II) is bioaccessible, or that only Fe(II) is bioaccessible. Irrespective of whether Fe(II) is bioaccessible or not, increasing its concentration increases the bioavailable pool of Fe by maintaining a higher concentration of Fe in the dissolved phase which can be (and is) actively cycled via Fe(III) phases.

Q. Line 67: Meaning. . . R. Line 67: We just wanted to frame this in an environmental context, this is a high fraction relative to what you can observe in the natural environment (much higher, for example, than the ratio in bulk seawater and comparable to that observed in a highly-photochemically affected context).

Q. Line 77: 2010 not 2013 R. Line 77: change done, 2010.

Q. Line 81: Why 'therefore'? The preceding sentence deals with a case that is not in a HNLC area of the ocean. R. Line 81: Sentence changed to another connector.

Q. Line 94: Perhaps in fresh water but I imagine any shift in seawater pH is extremely transient and localised due to strong buffering! R. Line 94. Yes, clarified, although the fjord surface layer is relatively fresh and thus more weakly buffered.

Q. Line 99: 2010 not 2013. R. Line 99: change done.

Q. Line 148: Had the ash been rained on in the interval between the eruption and sampling? R. Line 148: No, the ash was gathered from fresh deposition at the eruption nearby area, from ash than had been falling down on the day before.

Q. Line 151: But South Atlantic seawater is presumably not HNLC water, so its properties might lead to findings here only relevant to ash input to the South Atlantic. For example, HNLC seawater might have different types/abundances of Fe-binding ligands. R. Line 151: Yes, in short, the exact seawater used will have several influences on the Fe leached from ash int solution. Ligand concentration being one of them. This affects any leach experiment and is not specific to the work herein. S Atlantic water with a relatively typical dFe starting concentration was used as the most representative seawater supply we could access.

Q. Line 152: Is there a basis for this range? Does it mimic the Calbuco ash loading to the SW Atlantic and SE Pacific regions studied? R. Line 152: As we don't have data about the natural ash loading at the ocean regions studied here after the Calbuco eruption, we can't answer this question. The ash loadings are designed to be broadly comparable to the range used in prior work. There is no clear reason to choose a specific loading, as the loading gradient in any eruption is always very broad both

spatially and temporally.

Q. Line 167: Again, if South Atlantic seawater is not Fe-limited, then presumably measurements of Fe(II) concentrations in this water cannot be generalised to reflect Fe release behaviour from ash in Fe-limited seawater. For instance, different types/abundances of Fe-binding ligands in HLNC water might strongly influence dissolved Fe concentrations on ash input to seawater. R. Line 167: It's not clear what the reviewer means here, there is no such thing as Fe-limited seawater, Fe-limited refers to phytoplankton status. The ratio of dFe:macronutrients could be used to indicate if seawater is deficient. Generally higher ambient dFe concentrations (i.e. non-dFe-deficient conditions) would lead to saturation more easily, so the Fe-leached from ash would be potentially reduced compared to starting with lower ambient dFe concentrations. With respect to ligands, 'Fe-binding ligands' almost invariably refers to Fe(III)-binding ligands5, the effect of which would not be direct on short-term (seconds-1 minute) Fe(II) release into solution. Fe(II)-binding ligands may exist in an environmental context, but they are not thought to compose such a large influence on Fe(II) speciation as Fe(II) is a transient specie the distribution and concentration of which is dominated by redox dynamics rather than by ligand properties.

Q. Line 168: Why is the ash loading concentration different than that used for the trace metals above? R. Line 168: because these are two different experiments. For Fe(II), we noticed that there was pronounced sensitivity to the ash loading, so expanded the range of conditions to see what the very high/low loading/Fe(II) release looked like.

Q. Line 175: What 'dissolution experiments'? Do you mean ash leaching in seawater? R. Line 175: Yes.

Q. Line 179: Should be desert and please specify which, for consistency with specifying the ash source (Etna, Chaiten). R. Line 179: No longer present in manuscript.

Q. Line 180: Please specify sources R. Line 180: No longer present in manuscript.

[Figure]

Q. Line 183: Please revise heading - species responsible for alkalinity, ions, and nutrients are not mutually exclusive. R. Line 183: heading of point 2.3 changed to: 2.3 Ash samples – DI and brackish leaching experiments.

Q. Line 184: I thought it was Reloncavi Fjord? R. Line 184: This was the fieldsite. But consider that this fieldsite had experienced a huge deposition of ash, we suspected that collecting water which already had experienced a large ash exposure (and at the time of our ash collection still had a visibly high ash loading) would not be particularly insightful for leach experiments. We therefore collected water from an adjacent fjord to conducted our leaching experiments with.

Q. Line 187: Do you mean deionized water here? If so, please say this instead to avoid misinterpretation to mean environmental/fjord fresh water. R. Line 187: Suggestion accepted, now says DI water.

Q. Line 188: 0.18 g ash in 40 mL is 4.5 g/L or a 1:222 ash:water ratio. Where does this come from (it is not from Witham et al. 2005)? R. Line 188: The total amount of that sample was divided in order to have a good number of replicates. Following most of Witham et al., (2005) recommendations, point 5: 1; 2; 3; 4; 8 and 9. Note that although the total mass of unhydrated ash was limited, the ash/water ratio for DI leaching experiments was of the same order of magnitude as for Jones and Gislason 2008, considering the total flow through their reactor. Finally, it's the time scales that are probably most relevant, together with the selection of the finest size fraction.

Q. Line 200: Were saturation indices calculated for all species in solution? R. Line200: We ran Visual MINTEQ 3.1 for average the ionic composition of the 63 um size fraction (which had the highest yield of leachates) together with three scenarios of pH (5, 6 and 7) and Fe (4.5) – note that since these parameters were not measured for the freshwater experiments they are merely parameterized here based on plausible values (e.g. Fe should be within an order of magnitude of marine leaching following the results shown in Jones and Gislason 2008). The results are included in the Supplementary

material, and show highly undersaturated conditions for almost all potential minerals except for iron hydroxide species.

Q. Line 222: This should come after "at 3 depths". R. Line 222: Suggestion accepted.

Q. Line 328: Isn't it 4.5 g/L (0.18 g in 40 ml)? R. Line 328: Suggestion accepted.

Q. Line 335: This is probably just a surface area effect (i.e. smaller size particles for the same mass of ash release correspondingly more ions due to the greater surface area in contact with solution). Leaching not dissolution. R. Line 335: Suggestion accepted.

Q. Line 338: Because CaSO4 salts are not as soluble? R. Line 338: CaSO4 salts are secondary minerals, among an array of species that are, according to the charge balance and the results from various time steps, far from equilibrium.

Q. Lines 350-352: I don't see support for this statement. From Figure 7 (right) showing all replicates, the couple of high Fe release values per unit ash mass seem like outliers. In fact two of the lowest ash loadings exhibit among the lowest Fe release per unit mass. R. Lines 350-352: Following comments from another reviewer, we have better grouped the behavior of the metals considering the quality of the fit over the applied ash gradient, and the magnitude of the change in concentration compared to the initial concentrations. New lines 379-388 are therefore re-written, and p values are added to new Figure 6.

Q. Line 356: Be careful with these statements, there is a lot of overlap of error bars so any apparent increase may not be significant. R. Line 356: Suggestion accepted. We have added regression fits and p values to the plots showing changes in trace metal concentration with time. We also discuss which elements we cannot discuss meaningfully due to the limited change compared to background concentrations. We have modified this discussion to exclude elements where we cannot meaningfully determine trends (new lines 379-388).

Q. Line 370: Why this temperature and not room temperature, like the others? R. Line

370: Fe(II) is unstable at room temperature under oxic conditions (now explained in the text).

Q. Line 374: If not in the legend, at least here in the caption you should state what they are (desert dust, glacial flour, iceberg particles) etc. R. Line 374: No longer in manuscript following earlier comments.

Q. Lines 374 - 375: It seems that the two other volcanic ash (Etna, Chaiten) are included here to represent older ash samples, but if that's indeed the purpose, this is not a valid comparison because the different Fe chemistries (total Fe content, Fe redox speciation, and Fe mineralogy) in these samples are likely to be greater drivers of their Fe(II) release behaviours than the different ash ages. This must be acknowledged, or else I suggest removing the Etna and Chaiten ash from this study altogether. R. Lines 374 - 375: No longer in manuscript following earlier comments.

Q. Line 381: The Figure 7 y-axis reports Fe(II) release in nmol/g. Please be consistent for clarity. R. Line 381: We have shown the Fe(II) results as nmol g-1 in figure 7 because of the sharp curve that results, in Figure 6 we plot the concentration as nM because the propagated error on nmol g-1 becomes huge for those elements that do not show a pronounced change. As the data displayed as nmol g-1 was however also requested by another reviewer, we add it in the supplement.

Q. Line 391: Perhaps only at low ash loadings, it's hard to say from the few data points for 9 month old ash. I would not consider this a clear trend at all, and in fact the Fe(II) release from 4 month old ash is often higher than from 2 week old ash at the same ash loading. R. Line 391: No longer in manuscript following earlier comments.

Q. Line 393: This is not at all supported by the data shown in Figure 7 (left), see my comment above. R. Line 393: No longer in manuscript following earlier comments.

Q. Line 397: Are you sure that this corresponds to volcanic ash and not to volcanic sulphate aerosol? R. Line 397: The UV Aerosol Index largely reflects the strongly UV-

absorbing (dust) aerosols. Sulphates are weakly/non-absorbing aerosols and therefore thought to have a more restricted contribution to the absorption signal. Reference: Torres, O., Tanskanen, A., Veihelmann, B., Ahn, C., Braak, R., Bhartia, P.K., Veefkind, P. and Levelt, P., 2007. Aerosols and surface UV products from Ozone Monitoring Instrument observations: An overview. Journal of Geophysical Research: Atmospheres, 112(D24).

Q. Line 511: Again, have you tested for saturation of Fe(II) (and other species) in your leachates? It would be useful to explore the possibility of secondary phase precipitation explaining decreasing dissolved Fe(II) with increasing ash loading. R. Line 511: (Saturation is tested for in the Supplement). Specifically for Fe(II), Fe(II) is inherently unstable under these conditions, decaying on a timescale of minutes even with the reduced temperature. It therefore doesn't really make sense to us to consider the extent to which Fe(II) is saturated in solution. With respect to dFe, these concentrations are likely over-saturated as ligand concentrations in this same S Atlantic water have previously been determined to be about 1.5 nM.

Q. Line 512: Again, I don't see clear evidence for this statement in the data (e.g., Figure 7 right). R. Line 512: Please see comment above for line 393. We have re-written this section.

Q. Line 520: Although this notion is propagated in the literature, there remains a paucity of evidence for Fe salts on ash surfaces! This section is missing important information on the forms of Fe (Fe(II) and Fe(III) in ash) - in aluminosilicate glass and mineral network and Fe(-Ti) oxide minerals. Will mislead readers to claim Fe salts are responsible again when we know that's not the case. R. Line 520: As per above comment. Our Fe(II) vs time plots show relatively unambiguously that there was a sudden pulse of Fe(II) release into solution occurring between 0-60 s after ash addition. We do not think that the shape of the Fe(II) vs time plot is not consistent with leaching from a solid aluminosilicate glass phase and suspect it is more consistent with a sudden dissolution effect. However we recognize this is not unambiguous and accordingly have rewritten

(new lines 562-574)

Q. Lines 522 – 524: No. Fast release of Fe(II) is more likely to originate from leaching of the aluminosilicate glass. Please do your homework (e.g., see Maters et al. 2016 and 2017 - those studies done at pH 1, 2 and 5 but are still relevant sources of info on the forms of Fe in ash and its release into solution)! R. Lines 522 – 524: These leaches in acid cannot be extrapolated to seawater at pH 8 (see earlier comment). If it were the case that Fe(II) were released from a mineral phase, we do not think that this would produce the Fe(II) vs time distribution that we observed during the Fe(II) incubation experiments.

Q. Line 525: See earlier comments, this statement is simply not well supported by the available data. R. Line 525: No longer in the text following earlier comments.

Q. Line 527: There's no such thing as acidic surface coatings. The presence of any salts on ash surfaces is the end product of prior reaction between acids (H2SO4, HCl) and the aluminosilicate –> neutralization. R. Line 527: No longer in the manuscript following earlier comments about the temporal development of Fe(II)

Q. Line 529: How would an acid-base reaction be responsible for Fe(II) conversion to Fe(III)? Presumably the Fe(II) at the ash surface has somehow been oxidised to Fe(III) during storage, or else made to be less mobile in some other way... R. Line 529: No longer in the manuscript following earlier comments about the temporal development of Fe(II)

Q. Line 530: Again, based on the fact that your data do not support the conclusion that aged ash releases less Fe(II), this statement should be removed. In any case, the term 'aged' in the mineral dust/glacial flour/volcanic ash community often refers to material that has interacted with other species during atmospheric transport ('aging'), if anything increasing Fe solubility and Fe(II) mobilisation over time (e.g., see Maters et al. 2016). R. Line 530: No longer in the manuscript following earlier comments about the temporal development of Fe(II)

Q. Line 532: This is true simply because we know that airborne material (dust, flour, ash) undergoes atmospheric processing, including exposure to inorganic and organic acids and cloud condensation and evaporation cycles, that is likely to modify the Fe solubility and speciation in the material before deposition to water bodies. Please acknowledge the huge body of literature in this area. R. Line 532: we believe that in this statement the reviewer is referring to airborne material that has undergo atmospheric processing due to long transport time-distance. Here we worked with fresh ash that was deposited in the nearby volcano area shortly after its release, thus we are not concerned with processes occurring in the atmosphere days to months after an eruption. We were referring specifically to a method artefact – when conducting a dFe leach at room temperature, any Fe(II) released (or at least a substantial fraction of it) will precipitate before it can be measured as dFe using standard leaching techniques.

Q. Lines 536 – 544: All this should be removed because it surrounds a claim about aging/Fe(II) release that is not supported by the data here. R. Lines 534 – 544: No longer in the manuscript following earlier comments about the temporal development of Fe(II)

Q. Line 552: What aerosols? Please specify since the Fe chemistry in different particulate materials can vary drastically. R. Line 552: The chemistry of these aerosols was highly variable as it refers to a transect over a large area of the offshore Pacific (we clarify the wording in the text "from aerosols collected across zonal transects of the Pacific Ocean"

Q. Line 553: Specify Calbuco 2015 ash. Saying 'fresh volcanic ash' is a gross generalization and completely neglects existing studies reporting variable Fe release from ash, including Fe(II) and Fe(III) release by ash from different eruptions - Maters et al. 2017. Okay that study done at low pH but it shows that Fe chemistry in ash is highly variable and likely plays an important role in Fe release from ash in solution. R. Line 553: Changed to 'Calbuco' as suggested.

Q. Line 561: Please specify ash or particle, if that's indeed what the satellite detected. The term 'aerosol' in the volcanology context most often refers to sulphate aerosol. R. line 561: For clarity we have replaced 'atmospheric aerosol loading' with 'UV aerosol index'

---

## Author Comment (AC3) · 20 Nov 2020

Q Questions. R replies.

Q. Specific comments: The ash sample was collected two weeks after the eruption and the meteorological data in Figure 3 indicates some rainfall (albeit in a different location to where the ash was collected) in the period between eruption and sample collection. Is it possible that the collected ash had been exposed to rainfall before collection? If

so, can the authors discuss how this may have influenced their findings in terms of leachable trace elements and major ions?

R. To Specific Comments: the ash sampled was collected after the third eruptive pulse and the collected ash was from a dry surface that was not rained before sampling. This was done to decrease the uncertainty of following leaching procedures.

Q. 1): The leaching experiments conducted for determination of dissolved TMs into seawater, Fe(II) into seawater, and alkalinity and major ions into brackish and deionized water all use different experimental approaches, in terms of volumes used, ash loading, and length of mixing time. This is most relevant for the comparison of dissolved Fe versus Fe(II). While such differences in approach are sometimes unavoidable, the authors should at least discuss the potential for complications resulting from these differing approaches, particularly for the iron data – can they rule out any methodological artefacts in the data? R. 1) We did use different experimental approaches, but for different objectives and specific reasons. Regarding our most important finding on using fresh ash and the effect of Fe release, we did follow similar leaching protocols in order to avoid methodological artifacts of the data. The switch of method to a larger volume and colder temperature for Fe(II) measurements was done precisely to avoid artefacts in the Fe(II) data as at room temperature we would under-estimate any Fe(II) release, and with a low volume the constant removal of solution for flow injection analysis would potentially change the ash loading during the experiment. The specific potential issue with ash particles being measured as Fe(II) can be ruled out (see below) based on the absence of an increase in measured Fe(II) with increasing ash, and no detectable Fe(II) after experiments were conducted and blanks were run through the instrument.

Q. 2) The method for the Fe(II) release leaching experiment states that subsamples were introduced into the flow-injection system without filtration. Does this not admit a potential positive bias in the released Fe(II) data through small ash particles getting trapped in the FIA manifold and undergoing further leaching and/or reaction with the FIA reagents? R. 2) We tested for this, both the deionized water blanks run before/after

loading (which were always below detection) and the absence of an increase in Fe(II) signal with increased ash loading suggested that there was no detectable effect associated with ash particles running through the apparatus either being detected as what we assume is dissolved Fe(II), or 'sticking' and causing an increased Fe(II) signal. We should also note that the FIA inflow was rigidly positioned at mid-depth in the incubation bottles used to prevent potentially large particle uptake if/when ash settled from suspension.

Q. 3) In the Fe(II) method description it states that measurements were made every two minutes for 30 minutes for each ash loading, and that the data presented are "mean concentrations measured from 2-30 minutes after adding ash into solution". Does this mean that all of the data from 2-minute intervals are averaged to produce the data points in Figure 7? Was there no significant temporal progression of concentration over this 30-minute period? A related point is that if the data points in Figure 7 are mean values, presumably the standard deviations could be added to give a clearer idea of the significance of differences between datapoints. R. 3) Yes this is correct, and in hindsight we should have shown the temporal trend as this also provides some insight into the origin of this Fe(II) as discussed by reviewer 2. In order to discuss the time-series across all experiments together, we normalize each experiment (i.e. each ash incubation followed by >30 minutes of monitoring) to the median concentration of each individual time-series. The whole set of experiments can then be considered together, where 1.0 on the y axis corresponds to the median Fe(II) concentration observed, to see the general temporal trend (with standard deviations – these are not shown on the plot with all experiments for clarity).

Q. 4) Is there any scientific significance to the two ash size fractions chosen in the alkalinity/major ions leach experiments? R. 4) Yes, the two different ash size fractions were chosen to look at the effect of different particle size, as is known that leaching can vary drastically because smaller particle size has a much larger bigger surface for interactions related to its mass. Ultimately, based on the size fractions represented above,

we tested both the dominant size fraction by mass (250-1000 um) and by surface area (<63 um), the latter proving an overwhelming contribution to leaching products as demonstrated here. The size fractions of the ash are now stated for clarity (lines 215-217).

Q. 5) The description of dFe release in section 3.4 is described as being most efficient at the lowest ash loading per unit volume of seawater (line 351). Similarly, in Section 4.2, release of Fe and Mn is referred to in terms of nmol/g. This would be easier for the reader to visualize if Figure 6 was altered. Either additional plots could be included to plot each element as nmol/g released versus ash concentration in mg/L (as in Figure 7), or these plots could be superimposed on the existing plots by including secondary x- and y-axes. R. 5) Yes, this is now provided in the supplement.

Q. 6) For the Fe data in Figure 6, the value for the lowest ash addition has a large standard deviation. In Figure 7 we see that this is due to one replicate with a very high amount of Fe released per gram and two replicates with low values. The difference is very striking. Can the authors comment on the likelihood that the high value is an outlier and/or due to sample contamination? If not due to contamination, could this value be due to a methodological artefact? R. 6) Yes, looking at all the trace elements, there are some triplicates with relatively large standard deviations. This is not unique to Fe, and is generally more common for low concentrations of Pb, Cd, Co and Ni. There are several contributing factors to this; the deduction of the initial concentration adds to the uncertainty especially when the net change after ash addition is low, for the lowest ash loadings the number of particles is low and therefore any therefore variability between replicates likely poorer. The highest of the dFe values could be labelled an outlier, but so could several of the other individual measurements if we look at all metals. Although blanks and replicate measurements were always ok, contamination of an odd value is always possible, but it is difficult to conclusively separate this from the inherent variability in metal composition when using small quantities of ash. This raises an important point which we now use to better separate the different metals into groups – that we

should consider the measured change in solution relative to the background concentration and its variability (i.e. to the starting concentration and its standard deviation) as with some elements it is not possible to meaningfully discern trends from background variation.

Q. 7) At line 381 the release of Fe(II) from ash is referred to in terms of nmol/L, but Figure 7 does not show this as it relates to ash loading. It may be useful to include an additional panel in Figure 7 that shows the nM Fe(II) release as a function of ash added. R. 7) We can show the same data with different units, but prefer to do this in the supplement.

Q. 8) At line 393, the authors mention an apparent decrease in Fe(II) release with aging, but the only notable decrease seems to be between 4 months and 9 months, with little apparent difference between the trends at 2 weeks and 4 months. This should be clarified (e.g. "The release of Fe(II) from ash therefore appeared to decrease with aging after several months"). R. 8) No longer in the manuscript following earlier comments about the temporal development of Fe(II) from another reviewer

Q. Comment on section 3.2 - Figure 4.) In section 3.2, this figure is used to make a comparison between diatom abundance at stations in the upper part of the fjord, and historical data from Reloncaví Sound, which presumably undergoes more circulation and has a shorter residence time for waters. In addition, the new data is integrated over the upper 15m, compared to the upper 10m for the literature data. The comparison is quite striking. I appreciate that there is a desire to put the new data into some kind of historical context, but I think the authors should include the caveat in section 3.2 that the new and historical datasets may not be directly comparable. The authors do state in the discussion that the data is not directly comparable to the historical data (lines 426-427), but I think this point also needs to be made in the results section. R. Comment on section 3.2 - Figure 4.) Suggestion accepted. Now after the first sentence says "Historical data from González et al., (2010) is not directly comparable with recent data but gives a site-seasonal useful context." We calculate the potential difference in

our data if it were integrated to a different depth (10 m or 15 m), the difference is about 20%.

Q. Comment on Figure 8.) The apparent differing responses in the eastern Pacific and western Atlantic to ash deposition is a very interesting aspect of this study. However, I believe it would strengthen the findings of this paper if the authors could rule out the possibility that the observed response in chlorophyll in the Pacific Ocean is coincidental to the ash input. Figure 8G compares the satellite-derived aerosol index and chlorophyll-a concentration in the Pacific region over which the ash cloud passes. Have the authors looked at making a chlorophyll-a time-series at a similar area that did not see a strong variation in the aerosol index (for example, the area immediately to the south of the box used in 8G)? If no chlorophyll-a bloom corresponding to that in 8G is observed at this "no-ash" site, it would strengthen the argument that ash deposition was the trigger. Similarly, on the Atlantic side, there is a smaller ash-impacted area to the south of Rio de la Plata evident in panels C and D. Have the authors looked for any possible chlorophyll-a signal in that region and if they have, do the findings concur with the findings in panel H (i.e. that there is no ash-driven bloom)? Admittedly the aerosol index for this area looks substantially smaller, and cloud cover in the later time-period covered (panel F) may prevent a proper analysis of this area. Line 557 - The Browning et al (2015) reference suggests that in some cases, ash can bias satellite-derived chlorophyll-a measurements upwards significantly due to the optical properties of the ash and the algorithms used to convert data into chlorophyll concentrations. Can such a bias be ruled out in this study? R. Comment on Figure 8.) Browning et al. (2015) found that in very low chlorophyll-a cases, the presence of ash in seawater could bias satellite-detected chlorophyll-a concentrations upwards. This bias was found to strongly decrease with increasing chlorophyll-a, such that at concentrations >~0.5mg/m3 the calculated impact was very small (Fig. 4b in Browning et al., 2015). Background chlorophyll-a concentrations in the targeted satellite study regions were at or above these levels, suggesting the deposited ash would have relatively limited impact on satellite-retrieved chlorophyll-a concentrations.

We have now conducted a similar analysis for a control region of the Pacific and south of the Rio de la Plata.

"Experiments with ash suspensions have shown that ash loading has a restricted impact on satellite chlorophyll-a retrieval 6, therefore offering a means to assess the potential biological impact of the 2015 Calbuco eruption in offshore waters. We found evidence for fertilization of offshore Pacific seawaters in the studied area (Fig. 8). Following the eruption date, mean chlorophyll-a concentrations increased ~2.5 times over a broad region where elevated atmospheric aerosol loading was detected (Fig. 8G). Both the timing and location of this chlorophyll-a peak were consistent with ash fertilization, with the peak of elevated chlorophyll-a being located within the core of highest atmospheric aerosol loading, and the peak date occurring one day after the main passage of the atmospheric aerosol plume. A similar phytoplankton response timeframe was reported following ash deposition in the NE Pacific following the August 2008 Kasatochi eruption 7 which was similarly thought to be triggered by relief of Fe-limitation 8. At the same time, a control region to the south of the ash-impacted Pacific region showed no clear changes in chlorophyll-a matching that observed in the higher UV aerosol index region to the north.

A smaller ash impacted area to the south of the Rio de la Plata, where nitrate levels are expected to be higher than to the north, but with Fe levels also expected to be elevated due its location on the continental shelf, showed a chlorophyll-a peak 7 days after the UV aerosol peak (Sup. Fig. 1). However, this was not well constrained due to poor satellite coverage in the period after the eruption. Considering the dynamic spatial and temporal variation in chlorophyll within this coastal area, it is challenging to associate any change in chlorophyll specifically with ash arrival."

Q. Line 71. No need for hyphen in micronutrient R. Line 71. Suggestion accepted.

Q. Line 179. and throughout Supplementary Table 1 – replace "dessert" with "desert". R. Line 179. Suggestion accepted.

[Figure]

Q. Line 187. "fresh water" – use deionized water throughout. There is potential for this to be confused with river water. R. Line 187. Suggestion accepted.

Q. Line 194. change to "a reproducibility of <2 umol/kg" R. Line 194. Suggestion accepted.

Q. Lines 303 – 305. Figure 4 – It appears that the legend for Figure 4 is incorrect. It looks as though diatom abundance is greater at stations A and C, rather than B and C as stated in the text. The data in supplementary table 2 indicates that the figure is wrong, rather than the text. Based on the supplementary table, I would say that circles are station A, triangles are station B, and squares are station C. R. Lines 303 – 305. Suggestion accepted. Figure 4 it was showing incorrectly the stations diatom data. In old figure 4, circles were station A, triangles B, and squares C. New figure 4 has data corrected. Text was also modified in order to accurately show the changes (figure legend).

Q. Line 304 and 431. in both cases it is stated that diatom abundances were measured on 16th May, yet the supplementary table gives the date as 14th May. Which is it? R. Line 304 and 431. Suggestion accepted. Measurements were made on 14th May, like the supplementary table show.

Q. Lines 305-307. It would be more accurate to say that highest measured chlorophyll was on 30th April at a station close to station C. Based on Figure 4 it can't be said that concentrations decreased to much lower concentrations in June, as there isn't any data shown for June. R. Lines 305-307. Suggestion accepted. Modified text now says: "Diatom abundance integrated to 15 m depth peaked at Stations A and C around 14th May, with notably lower abundances at the more freshwater influenced station B (Fig. 4), that is at middle point between all 3 major rivers. Highest measured Chlorophyll-a concentrations was on 30th April at Station C, including two nearby measurements from Yevenes et al., (2019), then chlorophyll-a values declined to much lower concentrations in late May which is expected from patterns in regional primary production (González

et al., 2010)".

Q. Line 339. No need for "and" after NH4. R. Line 339. Suggestion accepted.

Q. Table 1. It states in the caption that all values are means. It would be more informative to also include standard deviations in the table if the data is from replicates – this would allow readers to assess whether changes observed with time are significant or due to noise in the measurements. Also, how is the detection limit arrived at? Is it 3x standard deviation of a blank? R. Note now Table 2. Please note also there were some unit errors in the data for the larger size fraction, the correct values are now shown (this does not change any trends or our interpretation). Literature values are also shown. SDs are quoted in addition to the means. Yes this is correct.

Q. Line 356. I'm not convinced that Ni shows that trend – only two additions seem to give a positive increase in Ni concentration, with one of those being the highest ash loading, and this gives a false impression that there is a positive trend. I would group Ni with Cd rather than Co and Pb. R. Line 356. We created a new grouping system for trace elements by first considering whether or not a trend could be discerned from background variability (i.e. is the change large compared to the mean and standard deviation of the starting water) and then considering the linearity of the trend with ash. A new paragraph is added (new lines 374-388).

Q. Figure 7. The "ash 9 months" data does not match that in the supplementary table, in that in the table all four data points are between 18-31.9 mg/L. with corresponding nmol/g values of 2 to 16. R. Figure 7. Yes, there was an error in the table now corrected.

Q. Line 439 – change to "ash deposition per unit area" R. Line 439. Suggestion accepted.

Q. Lines 509 – 510. It seems more appropriate here to refer to Figure 7 (right hand panel), as that shows the data in terms of nmol/g, as mentioned in the text, rather than the nmol/L change shown in Figure 6. R. Lines 509 – 510. The text here has changed

following other comments.

Q. Lines 549 – 550. is there a possibility here that small particulates could have contributed to the Fe(II) concentrations (as these samples were not filtered between ash addition and analysis)? See specific comment earlier in review. R. Lines 549 – 550. Not really, as we would expect to see an increase in Fe(II) with increasing ash load if that were the case. There was also no evidence for measurably increased Fe(II) from particles 'sticking' inside the apparatus (blanks before and after experiments were below detection). We expect, but cannot explicitly prove, that the measured Fe(II) is therefore dissolved.

Q. Line 605. rather than "correlation", which suggests a statistical relationship between the two parameters, I would suggest rephrasing this to something more general, such as "atmospheric ash loading was related to an increase in chlorophyll-a" (that is unless the authors can include a panel in figure 8 that does indeed show a correlation between satellite derived chl-a and aerosol index). Note also that the ash distribution shown by the aerosol index does not necessarily translate to "deposition" as stated here. R. Line 605. Suggestion accepted.

———————————————————

---

## Referee Report (RR1)

**"A mosaic of phytoplankton responses across Patagonia, the SE Pacific and SW Atlantic Ocean to ash deposition and trace metal release from the Calbuco 2015 volcanic eruption" by Vergara-Jara et al. (Second review)**

Vergara-Jara and co-workers' present study is based on collected samples of ash fallout from the 2015 Calbuco volcano emission. Their leaching experiments were performed on two different sets of samples: (i) South Atlantic seawater (bulk ash leaching for trace metals (Fe, Cd, Pb, Ni, Cu, Co, Mn) and Fe(II), and (ii) DI water and Aysen fjord water (size-fractionated ash samples; <63 μm and 250-1000 μm) for total alkalinity (Aysen fjord, DI water) and major ions (DI water). In brief, the authors did a multitude of different leaching methods in the present work and looking at the present work one more time, I feel that the study is not that focussed. I firmly believe that the present study would have benefitted from a more focussed approach from the authors regarding their leaching experiments.

One of the critical aspects that I am still not entirely convinced with is some of the parameters/protocols used for leaching experiments conducted in the present work. Different leaching experiments were done on a different set of water samples (S. Atlantic seawater, brackish water from the Aysen fjord, DI water), making the results non-comparable, especially with previous such studies (Jones and Gislason, 2008). The study was also largely focused on (biased towards) explaining the results of the trace metal leaching experiments done on S. Atlantic seawater. Although the authors performed their leaching experiments on different ash size fractions (<63 μm and 250-1000 μm) on brackish water and DI water, they could not extract any major conclusions out of these experiments. Also, the discussion section primarily focussed on the results of leaching experiments done with the S. Atlantic seawater. So, considering all this, I am not entirely sure why the authors performed these different ash size-fractionated (<63 μm and 250-1000 μm) leaching experiments on a different set of samples (brackish water from the Aysen fjord, DI water).

When last reviewed, I had some concerns on how (and why) the authors defined different ash size fractions: <63 μm (fine fraction) and 250-1000 μm (coarse fraction) for their leaching experiments. Although the authors have tried to justify that they have used different ash size fractions "following the recommendations of Witham et al., 2005", I, specifically, did not find any such recommendations on these particular size fractions by Witham et al., 2005. Further, in connections with their leaching experiments with the DI water, even though the authors have claimed that "DI experiments provides additional opportunity for comparison with similar studies (e.g., Jones and Gislason, 2008)", the authors have not utilized this opportunity by discussing these comparisons. The results from the earlier study of Jones and Gislason, 2008 have been simply provided in Table 2, without discussing whether the similarities are valid or not. Additionally, (i) the leaching experiments performed by Jones and Gislason, 2008 were on a different ash-size fraction (45–125 μm), and (ii) the leaching experiments of Jones and Gislason, 2008 were conducted in Teflon single pass plug flow-through reactors (different from the present work). In light of these points, I do not think that authors' comparisons on their DI leaching experiments with earlier results of Jones and Gislason, 2008 would be valid. In fact, for making their comparisons to be validated against an earlier study (Frogner et al., 2001), Jones and Gislason, 2008 used the same experimental methods (I am quoting it here: "To allow direct comparisons with previous work, we have attempted to apply the same methods as used in Frogner et al. (2001)" from Jones and Gislason, 2008). So, at least, if the authors of the present work have performed their leaching experiments with DI water with an intension to compare their results with previous findings, they should discuss any similarities (or, discrepancies) in these comparisons in more details

i.e. to answer: why do they expect their results to be any similar or different from previous studies? What about different size fractions used (in contrast to Jones and Gislason, 2008)? What about differences in leaching experiments?

I also have major concerns regarding the processing of the S. Atlantic seawater used for trace metal leach experiments. The authors have not mentioned whether the S. Atlantic seawater sample was filtered or unfiltered. Besides, the authors' protocols for seawater processing deviated significantly from Jones and Gislason, 2008, wherein much more robust protocols for seawater processing were followed. Upon collection, Jones and Gislason, 2008 processed their seawater samples by filtering (through 0.2 μm filter; to remove particular matter) and subsequent irradiation with UV light (to kill the remaining biota) before storing the samples in the dark. Such robust seawater preservation methods (before leaching experiments) were found to be missing in the present study.

Regarding the trace metal leaching the protocols with the collected ash, the authors added pre-weighed ash into 100 ml of S. Atlantic seawater and gently mixed the suspension for 10 minutes. How did the authors decide on this particular mixing time (10 minutes)? Did the authors change the suspension mixing durations to see the impact of ash-interaction with seawater for a reduced or prolonged interaction duration?

I am also not sure why some of the parameters were changes during the manuscript. E.g., although the coarser size fraction was defined as 250-1000 μm fraction in the earlier part of the manuscript (section 2.3, Table 1), the same coarser fraction is defined as >1.0 mm later in the manuscript (section 3.3, Fig. 5).

In view of the above considerations, I think that a significant amount of work is needed on the manuscript before making it suitable for publication with the "Ocean Science."

---

## Author Response (AR2)

**Answers to reviewer 1**

1) I think Figure 1 still needs some work. One of the previous reviewers recommended adding a scale bar to Figure 1C, which the authors say they have done, but I could not find it. By scale bar, I assume that the reviewer was referring to distance in km in addition to the latitude and longitude data that was already included. In addition, the Petrohue river does not appear to be labeled, while the other two rivers are labeled in different font sizes. And the symbol for the hydrological station is difficult to see in the figure. Perhaps a solid black triangle would be better. In the caption, the ash plume extent should be stated as "approximate" as it represents a full week of coverage for a dynamic feature.

**Answer: Changes have been made to the new figure 1. A scale bar (in km) has been placed in addition to the lat/long, all three major rivers are labeled at the same font size and the station symbols are now bolder against the background. In the description and text is stated that the ash plume extent is an approximate representation for a full week of coverage of a dynamic feature (new lines 139-140)**

2) I think the collection of the ash sample still needs more clarification. The eruption is described in lines 144-145 as taking place in two pulses on 22nd April and 23rd April. The ash sample was not collected until 6th May. In the response to reviewer 3 the authors state that the ash sample was collected after the third eruptive pulse. The timing of this third pulse should therefore be included in the eruption description to clarify to readers that there was not much time between deposition and sample collection. (The authors do mention an ash plume on 6th May in the caption of Figure 2, but details of this third pulse should also be mentioned in the text when discussing the eruption timing).

**Answer: at the beginning of section 2.2 the timing of the sample collection of ash is clarified. We extent the introduction to mention the smaller 3$^{rd}$ eruption and now refer clearly to the main/first eruption throughout the text except where explicitly referring to the smaller 3$^{rd}$ eruption (new lines 147-150).**

3) With the inclusion of Table 1, there should be some reference to it in the text, preferably before getting into details about the different leaching experiments. I also suggest splitting the water type (DI water, seawater, etc) and the measurements made (trace metals, alkalinity) into separate columns in Table 1.

**Answer: Table 1 was modified to show the information in a clearer way with new columns and fewer rows and is referred to through the methods section (e.g. lines 177, 215)**

4) There is still a problem with Figure 4 and/or Supplementary Table 2. The station-by-station data for diatoms in the figure do not match those in the table. According to Supp.Table 2, the circles in Figure 4 represent diatoms at Stn.A, the triangles Stn.B, and the squares Stn.C. This problem was brought up in the previous round of review and the authors stated that it had been addressed, so it may be that the updated version of the figure somehow didn't get included. It needs to be addressed, along with any associated changes needed in the text regarding diatom abundance at the different stations (e.g. Line 326).

**Answer: Sup. Table 2 was correct, we apologize we had mis-interpreted a prior comment and amended the description of fig 4 in the text rather than the legend on the figure – the error was in the figure, with A/B/C not labelled correctly. Figure 4 has been changed to display the correct labels.**

5) The supplementary material for this manuscript includes five tables and three figures and yet reference is only made to one supplementary figure in the text (four of the tables are mentioned with regards to data availability). It would make sense to include references to the appropriate supplementary materials in the main text.

**Answer: All tables and figures are now properly mentioned in the text.**

6) In section 4.3 the authors no longer discuss the SW Atlantic area marked on Figure 8; instead referring to the area further south and shown on the supplementary figure. I think the better approach here involves first discussing the two Pacific areas, as they have done, and then comparing the two Atlantic areas – the one in Figure 8 that did not show any change in chlorophyll-a and then the one further south where location on the shelf and lower satellite coverage complicate interpretation.

**Answer: Significant changes have been made to section 4.3. to highlight the two impacted areas and Fig. 8 with Sup. Fig 3 are now discussed together in a logical order as suggested (new lines 682-722)**

Technical comments:

Line 49 – The symbol for fluoride should be F-, not Fl-. This was brought up by Reviewer 2 previously. Also at Line 227 and Table 2.

**Answer:  We have corrected this through all the text.**

Lines 68-72 – Suggest changing this sentence to "A pulse of Fe(II) release upon addition of Calbuco ash to seawater made it an unusually efficient dissolved Fe source. The fraction of dissolved Fe released as Fe(II) from Calbuco ash (~18-38%) was roughly comparable to literature values for Fe released into seawater from aerosols collected over the Pacific Ocean following long range atmospheric transport".

**Answer: Suggestion accepted, now in current lines 68-72**.

Line 96 – change to "some freshwater bodies"

**Answer: Suggestion accepted, current line 96**

Line 104 – Remove the first "North Atlantic" reference from this sentence and change the year of the Eyjafjallajokull eruption to 2010 (noted in the previous round of reviews).

**Answer: Suggestion accepted, current line 104 now reads "In contrast to the 2010 Eyjafjallajökull plume over the North Atlantic"**

Lines 106 – The use of "largely" in this sentence is made redundant by the use of "predominantly". Suggest removing "largely".

**Answer: Suggestion accepted, current line 106 has "largely" removed.**

Line 115 – Suggest changing the las part of the sentence to "..and leaching experiments carried out on ash collected from the fjord region, to investigate the inorganic consequences of ash deposition to natural waters".

**Answer: Suggestion accepted. New sentence can be found on lines 114-116.**

Lines 124-128 – I'm confused as to why the Petrohue is identified as the major river of the system (line 125) when the Puelo is the largest in terms of flow. Perhaps the first sentence could be adjusted to "The Calbuco volcano is located in close proximity to Reloncavi Fjord".

**Answer: Problem resolved due to mistake on naming the Petrohue as the major river.  Now reads "The Calbuco volcano (Fig. 1) is located in a region with large freshwater reservoirs and in close proximity to Reloncaví Fjord."**

Lines 161-163 – Ash was collected on 6 May. How long after deposition was this? Also clarify that the weather was dry throughout the period between deposition and collection. At present it is ambiguous – just the day preceding ash collection.

**Answer: We clarify details concerning the ash collection which was whilst the 3$^{rd}$ eruption (small) was ongoing, that the ash collection site was 30 km away from the volcano and that the ash was collected over a 24 h time period (i.e. freshly deposited ash from the 3$^{rd}$ eruption). Yes, the weather was dry throughout this time period (the time period plasticware was left outside for, and the day prior to this) see new lines 147-150 and specifically 163** "On 6 May (2015, Cochamó, Chile, approximately 30 km from the volcano) after the third, and smallest, eruptive pulse of ash from the Calbuco volcano (Fig. 2, A), and with the volcano still emitting material, ash was collected using a plastic tray wrapped with plastic sheeting (40 × 94 cm).  The plasticware was left outside for 24 hours until sufficient ash (~500 g) was collected to provide a bulk sample. Ambient weather over the period of ash collection, and the preceding day, was dry (no precipitation)."

Line 176 – The range of concentrations selected was justified in comments to a reviewer as being broadly in line with previous studies. This justification could be included in the text for all readers.

**Answer: In response to this, and comments from another reviewer concerning best practices from prior work, we include Sup. Table 1 showing all prior seawater leach work, and introduce a new paragraph evaluating the best options concerning leaches to investigate Fe release under conditions that might mimic natural conditions. New lines 178-213**.

Line 178 – Specify that the addition was of concentrated HCl.

**Answer: Suggestion accepted. Line 233**

Line 181 – Give the (approximate) seawater temperature.

**Answer: Suggestion accepted (now line 237), now approximate temperature is shown with reference to Sup. Table 1.**

Lines 183-184 – Suggest changing to "For these experiments, a pre-weighed mass of was added to 250ml South Atlantic seawater and manually shaken for approximately one minute, using an expanded concentration range of 0.2-4000 mg L-1"

**Answer: Suggestion accepted. New lines 240-243.**

Line 196 – change to "flow-through".

**Answer: Suggestion accepted. Line 253**

Lines 215-217 – The inclusion of the size distribution of the ash is a good addition. In the response to reviewers, the authors also make the argument that the <63um size fraction has the greatest surface area and so the choice of size fractions includes both the dominant size fraction by mass (250-1000um) and the dominant size fraction by surface area (<63um). A couple of lines making this argument should also be added to the manuscript to clarify for the reader.

**Answer: Suggestion accepted. New lines 277-279 "The dominant size fraction by mass was thereby the 250-1000 μm fraction which was analyzed in addition to the finest fraction (<63 μm) with the greatest surface area to mass ratio."**

Line 227 – Change Fl- to F-.

**Answer: Corrected (new line 289).**

Lines 234-240 – There is unnecessary repetition here. The first list of parameters could be removed, giving a sentence that reads "High temporal resolution (hourly) in situ measurements were taken simultaneously in the Reloncaví fjord (Fig. 1 C, North Patagonia Oceanographic Buoy) at the surface and at 3 m depth, using two SAMI sensors that measured spectrophotometric CO2 and pH (DeGrandpre et al., 1995; Seidel et al., 2008) (Sunburst Sensors, LLC), and an SBE 37 MicroCAT CTD-ODO (SeaBird Electronics) for temperature, conductivity, depth and dissolved O2, as per Vergara-Jara et al., (2019)."

**Answer: Suggestion accepted, we have shortened this section (new lines 296-300) "High temporal resolution (hourly) in situ measurements were taken in the Reloncaví fjord (Fig. 1 C, North Patagonia Oceanographic Buoy) at 3 m depth using SAMI sensors that measured spectrophotometric $CO_2$ and pH (DeGrandpre et al., 1995; Seidel et al., 2008) (Sunburst Sensors, LLC), and an SBE 37 MicroCAT CTD-ODO (SeaBird Electronics) for temperature, conductivity, depth and dissolved $O_2$, as per Vergara-Jara et al., (2019)."**

Line 254 – "an" instead of "a"

**Answer: Suggestion accepted, line 316.**

Lines 259-262 – Reword as "Additionally, as part of a long-term monitoring program at station C (Fig. 1), chlorophyll-a samples were retained from 6 depths (1, 3, 5, 7, 10 and 15 m) on 6 occasions during March-May 2015. Chlorophyll-a was determined by fluorometry after filtering 250 ml of sampled water through GFF filters (Whatman), as per Welschmeyer (1994)."

**Answer: Suggestion accepted, lines 319-324.**

Line 266 – "…for comparison to prior reported data integrated to 10m…"

**Answer: Suggestion accepted, line 327.**

Line 267 – I don't think it is necessary to reference Figure 4 here, particularly as Figure 3 has not yet been referenced.

**Answer: Suggestion accepted, reference to Figure 4 removed.**

Line 275 – Similarly, there is no need to reference Figure 7 here, when earlier figures have not yet been referenced.

**Answer: Suggestion accepted, reference to Figure 7 removed.**

Line 327 – Is station B the more freshwater influenced? Station A is further upstream in the fjord and close to the second largest river flow input. Station C is located close to the largest river flow input. Is there data to back up station B being more freshwater influenced?

**Answer: Changes were done to correct this, because the wording was ambiguous. A is in the inner-fjord although C sits at the mouth of a river outflow, so the term 'freshwater influenced' was ambiguous as it. (new line number 390-399), we now refer to the 'innermost station' (A) for clarity.**

Lines 329-330 – Reword this to state that highest chlorophyll concentrations were measured close to Station C on that date (rather than at C, including two nearby measurements).

**Answer: Suggestion accepted, new line 391 "The highest measured chlorophyll-a concentrations were on 30 April at Station C,".**

Line 358 - Change Fl to F (also Table 2). Also, elsewhere in the manuscript all of these chemical species are written as ions, which is what is measured by the technique, and should also be written as such here for consistency.

**Answer: Suggestion accepted (new line 421).**

Line 366 – The authors responded to a reviewer comment about brackish water leaching results not being included in Table 2 because the higher original concentrations of major ions in brackish water make it difficult to observe changes. A statement to that effect could be added here for the general reader.

**Answer: Suggestion accepted. New lines 430-432 states: "Major ion analysis was only conducted in de-ionized water as no significant changes would be observable for most of these ions in brackish or saline waters under the same conditions."**

Lines 376-377 – In the methods the ash loading for these leaching experiments is described as 2-50 mg/L, and in Figure 6 it is given as mg ash added, but here it is described as 0.1-6 mg/L. I think this latter case is supposed to be mg/100mL or just mg added. Please change units or numbers for consistency.

**Answer: Suggestion accepted, numbers, units corrected "The net release of dissolved metals proceeded with varying relationships with ash loading over the applied gradient (2-50 mg L$^{-1}$)" (new lines 441).**

Lines 377-383 – No mention is made here of where Fe fits into this grouping.

**Answer: This section is expanded slightly to better discuss all elements, new lines 440-459.**

Figure 7 (left panel) – Now that only Calbuco ash is included in the figure, the x-axis label should be changed to "ash loading", rather than "dust loading".

**Answer: Suggestion accepted, X-axis label corrected as "ash loading / mg L-1".**

Figure 7 (left panel) – the lines fitted to each dataset (2 weeks, 4 months, 9 months) are not mentioned in the figure caption.

**Answer: Suggestion accepted. They are now mentioned explicitly in the caption (new line 474).**

Figure 7 – The caption (and description in the methods) indicates a 30-minute time-series of Fe(II) measurements, but the x-axis of the right-hand panel clearly extends to beyond an hour. Please clarify in the text and caption.

**Answer: Suggestion accepted, clarification introduced, now Figure 7 legend read as "Figure 7. Fe(II) release from Calbuco ash into seawater. Mean Fe(II) released into South Atlantic seawater over a 30 minute leach at 5-7°C (left). The same batch of Calbuco ash was subsampled and used to conduct experiments on 3 occasions after the 2015 eruption (2 weeks, 4 months and 9 months since ash collection). The lines are power law fits, with associated equations shown in the legend. The 3 time-series of Fe(II) concentrations following ash addition is considered collectively by normalizing the measured concentrations (right), such that 1.0 represents the median Fe(II) concentration measured in each experiment. All experiments were conducted for at least 30 minutes, those conducted with 4/9 months old ash were extended for 1 hour. The black line shows the mean response over 34 leach experiments with varying ash loading, the shaded area shows ± 1 standard deviation. The initial Fe(II) concentration (pre-ash addition at 0 s) in all cases was below detection and thus the detection limit is plotted at 0 s (open circle). Source data is provided in Supplementary Table 2.".**

Line 415 – "Concentrations of up to 4.0 nM Fe(II)" are not evident in Supplementary Figure 2 (which should be referenced) or in the data in Supplementary Table 1. Perhaps 4nM represents one value from replicates of the 4 month old ash, 2mg/L ash loading experiment (2.17+-1.07 nM Fe(II) in the table), but that should be acknowledged in the text.

**Answer: Suggestion accepted, yes the 4.0 nM refers to a measured concentration, but not a mean (which is what is displayed in Sup. Table 2). For clarity, we refer to the mean Fe(II) concentrations that were observed, new line 491 "Elevated Fe(II) concentrations (mean 0.8 nM, Sup. Table 2) were evident at this temperature (5-7°C), which represents an intermediate sea surface temperature for the high latitude ocean."**

Line 419 – reference Supplementary Figure 2 also.

**Answer: Suggestion accepted, reference placed e.g. new lines 237, 495.**

Line 431 – in response to a comment from Reviewer 2 the authors provide a reference (Torres et al 2007) to support their interpretation of UV Aerosol Index as a measure of dust/ash aerosol loading. Why not cite that reference here in the text?

**Answer: Suggestion accepted, reference placed at section 2.6, line 334.**

Lines 436-438 – reference the appropriate supplementary figure here.

**Answer: Suggestion accepted, reference placed, line 468.**

Figure 8 – dashed vertical lines (22nd April) in panels G and H are hard to discern. Perhaps red would be a more visible colour to use.

**Answer: Solid red lines are now used throughout to represent the eruption dates.**

Line 471 – change "ash fall" to "ash deposition event" to avoid confusion with the fall season, mentioned in the line above (or use autumn instead of fall for the season).

**Answer: Suggestion accepted, now instead of "ash fall" is "ash deposition event", throughout including new line 553.**

Line 491 – it is stated that the diatom bloom (peaking 14th May based on data available) occurred several weeks after the eruption. This adds to the confusion about collection of the ash sample, which was carried out a week earlier on 6th May. But in response to reviewers the authors state that this collection took place after a third eruptive pulse of ash. This suggests an additional ash deposition event closer to the bloom peak. Please clarify.

**Answer: As per previous comments, we clarified that a 3$^{rd}$ (small) eruption occurred after the two major eruptions. Specific to the above comment, new lines 561-564 "Peaks in diatom abundance were measured at two stations on 14 May one week after the third (small) eruptive pulse, and measured chlorophyll-a concentrations were highest close to Station C on 30 April (Fig. 4)".**

Line 544 – See my general point about changes needed to Figure 4 – currently the figure does not show a more modest diatom abundance at Station A, as stated here, but rather at Station B. Also, the authors again state that this date (14th May) is 3-weeks after the eruption – should it be three weeks after the main eruption (see line 491 comment)?

**Answer: As the figure 4 was labeled wrong with respect the correct station labels A/B/C, the data does show (Supp. table 3) that station C & B had higher diatom abundance than station A. As above comments instead of referring to an "eruption" we now clearly distinguish the major eruptions on 22/23 April from the later, small eruption on 30 April. (new line 629) "Considering the more modest peak in diatom abundance at the most strongly ash affected station (Station A, Fig. 4) and the timing of the peak diatom abundance 3 weeks after the main eruption,"**

Lines 570-574 – I believe this sentence is added based on a reviewer comment about previous work. In the response to reviewers the authors argue that these lower pH experiments may not be applicable at seawater pH and I think that caveat can be added to this sentence.

**Answer:  Clarified, new line 658- reads "Alternatively Fe(II) could be released from more crystalline Fe(II) phases. Prior work, at much lower pH (pH 1 H2SO4 representing conditions that ash surfaces may experience during atmospheric processing, but not in aquatic environments) suggests that short-term release of Fe(II) or Fe(III) is determined by the surface Fe(II)/Fe ratio which may differ from the bulk Fe(II)/Fe ratio due to plume processing (Maters et al., 2017).".**

**Answer to Reviewer 2**

Vergara-Jara and co-workers' present study is based on collected samples of ash fallout from the 2015 Calbuco volcano emission. Their leaching experiments were performed on two different sets of samples: (i) South Atlantic seawater (bulk ash leaching for trace metals (Fe, Cd, Pb, Ni, Cu, Co, Mn) and Fe(II), and (ii) DI water and Aysen fjord water (sizefractionated ash samples; <63 μm and 250-1000 μm) for total alkalinity (Aysen fjord, DI water) and major ions (DI water). In brief, the authors did a multitude of different leaching methods in the present work and looking at the present work one more time, I feel that the study is not that focussed. I firmly believe that the present study would have benefitted from a more focussed approach from the authors regarding their leaching experiments.

One of the critical aspects that I am still not entirely convinced with is some of the parameters/protocols used for leaching experiments conducted in the present work. Different leaching experiments were done on a different set of water samples (S. Atlantic seawater, brackish water from the Aysen fjord, DI water), making the results non-comparable, especially with previous such studies (Jones and Gislason, 2008).

**Reply: This is a generic criticism that affects all work conducted on volcanic ash, not just this study. There are only 2 leaching experiments that can be done on water in a completely comparable way; synthetic seawater, or de-ionized water. These have no environmental relevance at all and are particularly problematic for Fe. The objective of this study was not to compare ash to other eruptions, but to understand environmental changes in marine environments around the region where ash was deposited.**

**Fe(II) leaches in deionized water, or synthetic seawater at room temperature would be meaningless as the features of natural water that moderate Fe(II) release are not replicated in either case, so we disagree with this comment. The details of the water selected for any leach should be matched to the hypothesis. If the experiments were designed to compare specific details of inorganic chemistry contrasting ashes from different volcanoes, yes we agree a single standardized synthetic seawater leach would be appropriate, but that is not what we were doing herein.**

**To clarify to what extent our work is/is not comparable to prior work we include a new paragraph describing the rationale, and the set up of prior work which investigated ash additions to seawater (new lines 172 - 214).**

The study was also largely focused on (biased towards) explaining the results of the trace metal leaching experiments done on S. Atlantic seawater.

**Reply: This is the main (possibly only) mechanism that could explain an increase in offshore primary production, as was detected by satellite derived chlorophyll (Fig. 8) so yes this is of course a focus of the text. We now explain this rationale in lines 172-214.**

Although the authors performed their leaching experiments on different ash size fractions (<63 μm and 250-1000 μm) on brackish water and DI water, they could not extract any major conclusions out of these experiments. Also, the discussion section primarily focussed on the results of leaching experiments done with the S. Atlantic seawater. So, considering all this, I am not entirely sure why the authors performed these different ash size-fractionated (<63 μm and 250-1000 μm) leaching experiments on a different set of samples (brackish water from the Aysen fjord, DI water).

**Reply: As stated in the introduction (new lines 172-214), we want to investigate to what extent the ash could affect primary production over the spatial scale here the ash was deposited which covers a**

range from fresh, to estuarine, to offshore waters. Hence the main mechanisms that could plausibly affect primary production in these waters (release of bioavailable N, Si, Fe, Mn; moderation of carbonate chemistry) should be investigated in conditions that represent these waters. Changes in the concentrations of ions and changes in the carbonate system could only really potentially affect primary production in freshwater (or perhaps brackish water) systems, as the reviewer pointed out previously. Conversely, trace metal fertilization could only really positively affect primary production to a large extent in offshore marine systems. So the experiments are designed to quantify these potential shifts in a meaningful way. Ionic leaches in seawater would be pointless, and conversely trace metal leaches in deionized water to quantify Fe release in seawater would be of dubious environmental relevance.

When last reviewed, I had some concerns on how (and why) the authors defined different ash size fractions: <63 μm (fine fraction) and 250-1000 μm (coarse fraction) for their leaching experiments. Although the authors have tried to justify that they have used different ash size fractions "following the recommendations of Witham et al., 2005", I, specifically, did not find any such recommendations on these particular size fractions by Witham et al., 2005. Further, in connections with their leaching experiments with the DI water, even though the authors have claimed that "DI experiments provides additional opportunity for comparison with similar studies (e.g., Jones and Gislason, 2008)", the authors have not utilized this opportunity by discussing these comparisons.

**Reply: Prior literature does not specify which exact size fractions should be used, merely guidelines concerning the consideration of size fraction as an important consideration. Our rationale for these specific size fractions is now stated in lines (278-280) "The dominant size fraction by mass was thereby the 250-1000 μm fraction which was analyzed in addition to the finest fraction (<63 μm) with the greatest surface area to mass ratio".**

The results from the earlier study of Jones and Gislason, 2008 have been simply provided in Table 2, without discussing whether the similarities are valid or not. Additionally, (i) the leaching experiments performed by Jones and Gislason, 2008 were on a different ash-size fraction (45–125 μm), and (ii) the leaching experiments of Jones and Gislason, 2008 were conducted in Teflon single pass plug flowthrough reactors (different from the present work). In light of these points, I do not think that authors' comparisons on their DI leaching experiments with earlier results of Jones and Gislason, 2008 would be valid. In fact, for making their comparisons to be validated against an earlier study (Frogner et al., 2001), Jones and Gislason, 2008 used the same experimental methods (I am quoting it here: "To allow direct comparisons with previous work, we have attempted to apply the same methods as used in Frogner et al. (2001)" from Jones and Gislason, 2008). So, at least, if the authors of the present work have performed their leaching experiments with DI water with an intension to compare their results with previous findings, they should discuss any similarities (or, discrepancies) in these comparisons in more details

i.e. to answer: why do they expect their results to be any similar or different from previous studies? What about different size fractions used (in contrast to Jones and Gislason, 2008)? What about differences in leaching experiments?

**Reply: As stated in the introduction, the main aim herein is to investigate the environmental effect of the Calbuco eruption, not to conduct an extensive comparison to every other eruption for which ash is available. Several extensive reviews have been written on this subject going into detail about what is, and is not, known concerning the effects of size, time, age, leachate on measured ionic and trace metal leaches which are cited extensively throughout. We do not think it is appropriate to repeat this discussion herein as it is ancillary to the main aim of the paper, particularly concerning the potential**

**effects of ash (change in alkalinity, and ion availability) that it is obvious from the results did not have any environmental effects on phytoplankton following the Calbuco eruption. Nevertheless, we have expanded the discussion of to what extent a comparison is/is not valid for the leaches (trace metal) that potentially did have environmental effects. See new Supp. Table 1 and new lines 172-214.**

**There is a very long discussion to be had concerning the precise design of any experiment to investigate any aerosol dissolution in seawater which we do not think is useful to discuss extensively herein, a general comment on the difference between the continuous plug flow reactor used by previous researchers and mixed reactor (bottle experiments) used here for example, is that flow through systems used a filter to retain sediments in place, which is susceptible to clogging, and hence limits its use to larger particle sizes which will not clog the system (the problem of clogging is something alluded to in the methods of Jones and Gislasson). As a consequence, previous or otherwise standard methods may be inadequate for assessing the effects of the most relevant fine particle sizes, which are generally more abundant and are certainly more widely deposited over larger geographic areas. Ultimately, residence time (and not reactor design) is the basis for the limited comparison between this experiment with previous experiments. Meanwhile, we can say that residence time was very similar compared to previous studies, and the concentrations and rates were within the same range.**

I also have major concerns regarding the processing of the S. Atlantic seawater used for trace metal leach experiments. The authors have not mentioned whether the S. Atlantic seawater sample was filtered or unfiltered. Besides, the authors' protocols for seawater processing deviated significantly from Jones and Gislason, 2008, wherein much more robust protocols for seawater processing were followed. Upon collection, Jones and Gislason, 2008 processed their seawater samples by filtering (through 0.2 $\mu$m filter; to remove particular matter) and subsequent irradiation with UV light (to kill the remaining biota) before storing the samples in the dark. Such robust seawater preservation methods (before leaching experiments) were found to be missing in the present study.

**Reply: For clarity we now include details of all prior work conducting ash leaches in seawater (Supp Table 1 and new lines 179-206). We disagree that the 'robust' technique above would have been appropriate to use herein for two reasons.**

**UV irradiation of seawater will destroy any natural organics present, it will also lead to high concentrations of compounds like H2O2 which will take weeks to months to decay back to background concentrations. The chemistry of Fe in seawater is controlled almost entirely by the concentration and nature of the organics present – organics which the reviewer previously commented we should think carefully about. There are of course circumstances under which it would be desirable to UV-treat seawater, but we do not think this is one of them as the cycling of Fe and Fe(II) would be interfered with.**

**The Atlantic water used was filtered (new lines 222), it was oligotrophic to start with, and the concentrations of any bacteria present after storage in the dark for > 1 year would be insufficient to make meaningful differences during leach durations of 10 minutes (this would apply even if water were unfiltered and leaches were conducted immediately for such a short leach duration).**

Regarding the trace metal leaching the protocols with the collected ash, the authors added pre-weighed ash into 100 ml of S. Atlantic seawater and gently mixed the suspension for 10 minutes. How did the authors decide on this particular mixing time (10 minutes)? Did the authors change the suspension mixing durations to see the impact of ash-interaction with seawater for a reduced or prolonged interaction duration?

**Timings vary widely in prior work, but all studies show that the major processes occur practically instantaneously such that there is little difference between initial leach timings of 10 minutes, 1 hour or 2 hours (see new lines 210-214) "A short leaching time (10 minutes + filtration) was adopted to minimize bottle effects and recognising that most prior work suggests a large fraction of Fe release occurs on short timescales (minutes), followed by more gradual changes on timescales of hours to days (Duggen et al., 2007; Frogner et al., 2001; Jones and Gislason, 2008)."**

I am also not sure why some of the parameters were changes during the manuscript. E.g., although the coarser size fraction was defined as 250-1000 $\mu$m fraction in the earlier part of the manuscript (section 2.3, Table 1), the same coarser fraction is defined as >1.0 mm later in the manuscript (section 3.3, Fig. 5).

**Answer: We apologize, there are only size fractions considered "<63" and "250-1000", the later was incorrectly referred to as ">1000" in one place in the prior work.**

In view of the above considerations, I think that a significant amount of work is needed on the manuscript before making it suitable for publication with the "Ocean Science."

---

## Author Response (AR3)

**Replies to editor (bold)**

1. Although the reviewers did not point it out, I found the authors are assuming that primary production and chlorophyll are the same. Primary production is not estimated but discussed quite a lot – in fact one of the objective of the study was this:

"We thereby evaluate the potential positive and negative effects of ash from the 2015 Calbuco eruption on marine primary production" (Lines 116-117)

and it is concluded: "Strong evidence of a broad-scale 'bottom-up' fertilization effect of ash on primary production was not found locally within Reloncaví Fjord," (lines 735-736)

I would suggest the authors to restrict their discussion to chlorophyll only in the absence of primary production data.

**We have checked this throughout and there were two places where we used the term 'primary production' too loosely, now amended as follows:**

**Line 117 "We thereby evaluate the potential positive and negative effects of ash from the 2015 Calbuco eruption on marine phytoplankton"**

**Line 741 "Strong evidence of a broad-scale 'bottom-up' fertilization effect of ash on phytoplankton was"**

2. How were diatoms enumerated? Should be explained in methods.

We expand the description of this as requested (line 314-). **"During May 2015, weekly field campaigns were undertaken in the Reloncaví Fjord. Phytoplankton samples were collected at 3 depths (1, 5 and 10 m) for taxonomic characterization and abundance determination at 3 stations (A, B and C; Fig. 1) using a 5 L Go-Flo bottle. Samples for cell-counts were stored in clear plastic bottles (300 mL) and preserved in a Lugol iodine solution. From each sample, a 10 mL subsample was placed in a sedimentation chamber and left to settle for 16 hr. The complete chamber bottom was scanned at 200× to enumerate the organisms and the result was expressed as number of phytoplankton cells per L of seawater (Hasle, 1978). Phytoplankton were identified to genus or species level, when possible, and divided into diatoms and dinoflagellates. Samples were analyzed using an Olympus CKX41 inverted phase contrast microscope and the Utermöhl method (Utermöhl, 1958). The phytoplankton community composition was then statistically analyzed in R (RStudio V 1.2.5033) using general linear models in order to find statistically significant differences between dates and group abundances."**

In addition, please edit the following:

Table 1: No. in "No of replicates" should be corrected.

Put caption on the top of the tables (Table 2)

**Table 1 "Number of replicates"**

**Table 2 Re-arranged.**